# A supramolecular lanthanide separation approach based on multivalent cooperative enhancement of metal ion selectivity

Xiao-Zhen Li[1,2], Li-Peng Zhou[1], Liang-Liang Yan[1], Ya-Min Dong[3], Zhuan-Ling Bai[4], Xiao-Qi Sun[1,3], Juan Diwu[4], Shuao Wang[4], Jean-Claude Bünzli[5] & Qing-Fu Sun [1]

Multivalent cooperativity plays an important role in the supramolecular self-assembly process. Herein, we report a remarkable cooperative enhancement of both structural integrity and metal ion selectivity on metal-organic $M_4L_4$ tetrahedral cages self-assembled from a tris-tridentate ligand ($L^1$) with a variety of metal ions spanning across the periodic table, including alkaline earth ($Ca^{II}$), transition ($Cd^{II}$), and all the lanthanide ($Ln^{III}$) metal ions. All these $M_4L^1_4$ cages are stable to excess metal ions and ligands, which is in sharp contrast with the tridentate ($L^2$) ligand and bis-tridentate ($L^3$) ligand bearing the same coordination motif as $L^1$. Moreover, high-precision metal ion self-sorting is observed during the mixed-metal self-assembly of tetrahedral $M_4L_4$ cages, but not on the $M_2L_3$ counterparts. Based on the strong cooperative metal ion self-recognition behavior of $M_4L_4$ cages, a supramolecular approach to lanthanide separation is demonstrated, offering a new design principle of next-generation extractants for highly efficient lanthanide separation.

[1] State Key Laboratory of Structural Chemistry, Fujian Institute of Research on the Structure of Matter, Chinese Academy of Sciences, Fuzhou 350002, People's Republic of China. [2] University of Chinese Academy of Sciences, Beijing 100049, People's Republic of China. [3] Xiamen Institute of Rare Earth Materials, Haixi Institute, Chinese Academy of Sciences, Xiamen 361021, People's Republic of China. [4] School for Radiological and Interdisciplinary Sciences (RAD-X) and Collaborative Innovation Center of Radiation Medicine of Jiangsu Higher Education Institutions, Soochow University, Suzhou 215123, People's Republic of China. [5] Institute of Chemical Sciences and Engineering, Swiss Federal Institute of Technology, 1015 Lausanne, Switzerland. Correspondence and requests for materials should be addressed to J.-C.Bün. (email: jean-claude.bunzli@epfl.ch) or to Q.-F.S. (email: qfsun@fjirsm.ac.cn)

anthanides, owing to their peculiar electronic structure, are used in a wealth of important applications, including batteries, display devices, contrast agents, magnetic or superconducting materials, and catalytic converters[1]. Because they tend to be distributed in relatively small concentrations and highly insoluble in their pure forms, solvent extraction has been the primary means to separate, purify and recycle lanthanides. Especially, separation from calcium, one of the primary associated elements in lanthanide minerals, such as cerite and loparite-Ce, with an ionic radius similar to those in the middle of the lanthanide series, is crucial in the refining of high-grade lanthanide concentrates[2]. Moreover, nuclear reactors generate a wide variety of waste products including radioactive mixtures, lanthanides, and transition metals, which also require effective conversion and separation for long-term storage and recycling[3]. Owing to the similar ionic radii and coordination numbers/geometries of the rare earth elements, as well as calcium and cadmium metal ions, traditional extraction processes must be conducted in a cascade for complete separation and purification of the desired element[4–6]. New separation techniques that eliminate processing steps and waste are, therefore, of great importance with respect to economic and environmental concerns[7–10].

Selective binding of metal ions has been a classical research topic in supramolecular chemistry since the very beginning of the field[11–14]. Covalent crown- or cryptand-based receptors possessing a complementary binding pocket are known to have specific ion-recognition properties[15–17]. Meanwhile, designed metal ion hosts that utilize the preorganization effect of the supramolecular scaffold have also been widely studied[18,19]. Moreover, a newly proposed solvent-free extraction/separation process is making use of resins derivatized with macrocyclic ligands[20]. However, almost all of these traditional extractants rely on the selective formation of mononuclear complex from metal ion mixtures. Metal selectivity in multinuclear complexes, especially for lanthanides, has been overlooked for a long time, possibly as a result of the limited number of stable multinuclear lanthanide complexes that exist in solution[21–24]. To the best of our knowledge, there are only a few examples of the control of lanthanide metal selectivity using self-assembly in the literature, and they are mainly based on the formation of dinuclear lanthanide helicate architectures[25–27].

Multivalency plays an essential role, both in the mediation of biological processes as well as in the construction of supramolecular structures[28–31]. An important example of a multivalent interaction in nature is the interaction between a virus and its host cell, which leads to a stable initial adhesion. Remarkable enhancement in stability has also been proven for the self-assembly of three-dimensional architectures and capsules through the cooperative effect of a vast amount of noncovalent interactions[32–36]. This strong multivalent cooperativity is believed to be the main driving force for the self-sorting phenomena observed in these multi-component assemblies, which bias the complicated system toward the selective formation of one well-defined structure[37–42]. However, until now the term self-sorting has been predominantly used when referring to the organic components in a metal–organic assembly[43–47]. In clear contrast, metal ion self-sorting has scarcely been studied, especially in the cases where ionic radii discrepancy is the only variable[48–50].

In our previous work, the first stereoselective self-assembly of chiral lanthanide tetrahedral cages was accomplished[51] with intriguing ligand self-sorting behavior owing to the strong supramolecular cooperative mechanical-coupling effect[52,53]. Herein, we report the unprecedented self-assembly capacity of $L^1$ with metal ions spanning across the periodic table, including alkaline earth ($Ca^{II}$), transition ($Cd^{II}$), and all the lanthanide ($Ln^{III}$) metal ions (M), ascribed to the appropriate rigidity of the $C_3$ symmetrical scaffolding, high assembly adaptability and

adequate chelating affinity with lanthanide ions of the neutral coordination motif. More importantly, this versatile ligand displayed rare and rather high discrimination between metal ions with identical coordination geometries as well as extremely small ionic difference, arising from supramolecular multivalent cooperativity, resulting in absolute or highly efficient metal ion self-recognition during mixed-metal self-assembly process. During both one-pot mixed-metal self-assembly and post-synthetic metal-metathesis experiments, we have observed unprecedented discrimination in favor of including the smaller lanthanide ions in the tetranuclear complexes. For comparison purpose, monodentate ($L^2$) and bis-tridentate ($L^3$) ligands, bearing the same coordination motif as $L^1$, have been synthesized; they turned out to be either unable to self-assemble, or form very fragile mononuclear and dinuclear complexes and no analogous high-precision metal ion self-recognition as that in the mixed-metal self-assembly of tetrahedral $M_4L_4$ cages was observed for the $ML_3$ or $M_2L_3$ counterparts. Furthermore, lanthanide extraction separation experiments have also been conducted using alkyl-functionalized ligands, taking advantage of the strongly cooperative metal ion self-recognition behavior of the tetrahedral cages.

## Results

**Self-assembly of $M_4L_4$ cages from $L^1$ with various metal ions.** When $L^1$ (8 μmol) was treated with $Ca(CF_3SO_3)_2$ (8 μmol) in $CD_3CN$ (500 μL) at 40 °C for 1 h, the quantitative formation of a

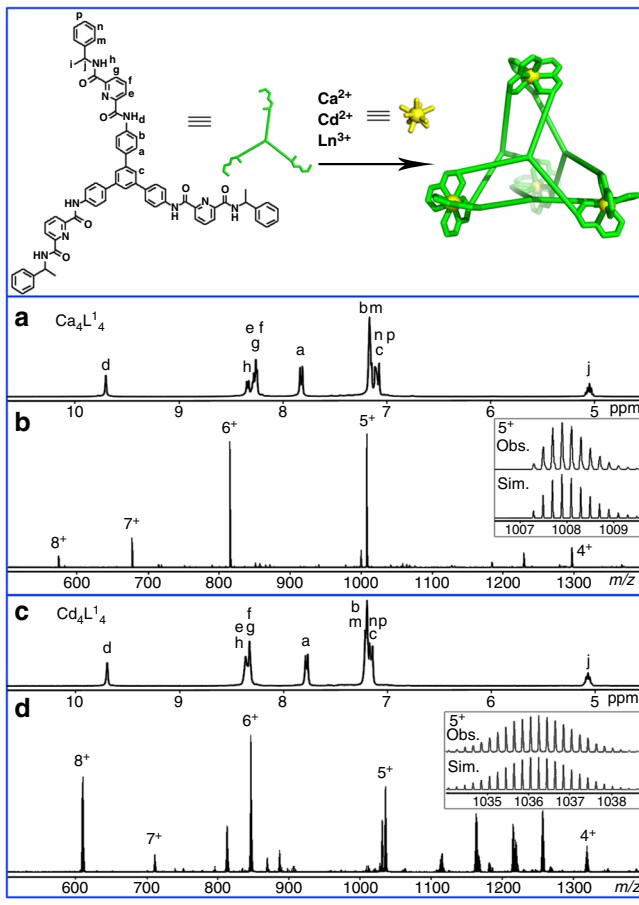

**Fig. 1** Schematic representation and characterization of the self-assembled $M_4(L^1)_4$ complexes. $^1H$ NMR spectra (400 MHz, $CD_3CN$, 298 K) and ESI-TOF-MS spectra of **a**, **b** $[Ca_4(L^1)_4](CF_3SO_3)_8$, **c**, **d** $[Cd_4(L^1)_4](ClO_4)_8$ with insets showing the observed (Obs.) and simulated (Sim.) isotope patterns of the 5+ peaks

single species was first confirmed by [1]H NMR (nuclear magnetic resonance) spectroscopy, where a single set of signals was observed, pointing to the equivalence of the ligand strands in the complex (Fig. 1a). The high symmetry of the product is further suggested by the [1]H-[1]H COSY spectrum, which shows the ligands experiencing identical magnetic environments (Supplementary Fig. 30). Furthermore, [1]H diffusion ordered spectroscopy (DOSY) shows that all the protons of the tetrahedral cages have the same diffusion coefficient, with a dynamic radii calculated with Stokes-Einstein equation to be about 14.5 Å (Supplementary Fig. 63), which is in good agreement with the reported structure of the $Eu_4(L^1)_4$ cage[51]. High-resolution electrospray ionization time-of-flight mass spectrometry (ESI-TOF-MS) analyses further confirmed the chemical formula of the isolated tetrahedral cage to be $[Ca_4(L^1)_4](CF_3SO_3)_8$, as shown in Fig. 1b. The spectrum features a series of peaks corresponding to multi-charged species with progressive loss of anions: for instance, peaks with $m/z$ equal to 677.3736, 815.0914, and 1007.9010 could be assigned to the charged molecular $\{[Ca_4(L^1)_4](CF_3SO_3)_n\}^{(8-n)+}$ complexes with $n = 1$ (7 +), 2 (6 +), and 3 (5 +). The assignments were also verified by carefully comparing the simulated isotopic distributions of the peaks with high-resolution experimental data.

Similarly, 1D and 2D NMR, ESI-TOF-MS confirmed the formation of a transition metal $Cd_4(L^1)_4$ cage under the same reaction conditions by replacing the metal source with $Cd(ClO_4)_2 \cdot 6H_2O$ (Fig. 1c, d, Supplementary Figs. 32, 64, and 210). The structure of this tetrahedral complex was unambiguously confirmed by single-crystal X-ray diffraction studies. Crystals of $Cd_4(L^1)_4(ClO_4)_8$ were obtained by slow vapor diffusion of dichloromethane into an acetonitrile solution of the complex. The X-ray structure of $Cd_4(L^1)_4$ shares most of the common features as that of the known $Eu_4L^1_4$ tetrahedral complex, except for crystallizing in a different $P6_322$ space group (Fig. 2, Supplementary Data 1, Supplementary Figs. 1–3 and Supplementary Table 7). It is worth pointing out that in this case all $Cd^{II}$ centers adopt nine-coordinating tricapped trigonal prismatic geometry. This is clearly different from the known $Cd_4L_6$-type cage assembled from another tris-tridentate ligand reported by Rizzuto et al.[54] recently, where the $Cd^{II}$ adopted a six-coordinating octahedral geometry. So the crystal structure of our $Cd_4(L^1)_4$ cage represents the first example of discrete supramolecular framework using nine-coordinated $Cd^{II}$ as vertices.

In similar reaction conditions as above, isostructural $Ln_4(L^1)_4$ tetrahedral complexes were also obtained by reaction of $L^1$ with different $Ln^{III}$ metal salts, as confirmed by 1D and 2D NMR,

spectroscopy (for $La^{III}$, $Ce^{III}$, $Pr^{III}$, $Nd^{III}$, $Sm^{III}$, $Yb^{III}$, $Lu^{III}$, and $Y^{III}$) and ESI-TOF-MS. (Supplementary Figs. 35–72, 211–219). The self-assembled complexes of diamagnetic $Y^{III}$, $La^{III}$, $Lu^{III}$ and weakly paramagnetic $Sm^{III}$ show small chemical shifts compared to those of the ligands, while paramagnetic $Ln^{III}$ ions ($Ce^{III}$, $Pr^{III}$, $Nd^{III}$, $Eu^{III}$, $Yb^{III}$) shift the complexes resonances downfield and upfield obviously (Supplementary Table 1 and Fig. 62).

Moreover, single-crystal X-ray diffraction analysis of $La_4(L^1)_4$ confirmed the tetrahedral molecular structure arrangement, consistent and isostructural with the reported $Eu_4(L^1)_4$ cage (Supplementary Data 2, Supplementary Figs. 4–6 and Supplementary Table 8)[51]. Although the discrete $M_4(L^1)_4$-type tetrahedral complexes are "isostructural" in nature, there are some distinct differences between the $Cd^{II}$ complex and the $La^{III}$ complex in their packing diagrams in the crystal states. $Cd_4(L^1)_4$ tetrahedral cages are very loosely packed in the ab plane and there are infinite channels with diameters of ca. 2.24 nm along the c axis. As a result, "void" occupancy as much as 50.6% is calculated in the unit cell based on PLATON[55]. In clear contrast, $La_4(L^1)_4$ tetrahedral cages are much densely packed and only 32.5% "void" occupancy is found by PLATON (Supplementary Fig. 7)[55]. As a result, crystals of $Cd_4(L^1)_4$ diffract much more weakly than those of $La_4(L^1)_4$.

The formation of $M_4L_4$ cages with various metal sources was inspiring, considering the facts that, until now, there was only one known discrete tetrahedral structure ($Mg_4L_4$) made from $AE^{II}$ (alkaline earth metal ions), which was self-assembled from anionic 1,3-dicarbonyl ligands[56]; and transition metal $Cd^{II}$ has been known to form only $M_6L_4$-type cages with tris-tridentate ligands[54]; and furthermore, in general, both $AE^{II}$ and $TM^{II}$ (transition metal ions) favor smaller coordination numbers instead of nine-coordinating tricapped trigonal prismatic geometry in their supramolecular coordination compounds[57–59].

**High-precision metal ion selectivity in $M_4L_4$.** Given the high self-assembly versatility of the tris-tridentate ligand ($L^1$), metal ion selectivity was examined through mixed-metal self-assembly experiments ($Ln_a^{III}/Ln_b^{III}/L^1 = 1/1/1$) and high-precision metal ion self-sorting behavior was observed due to the multivalent cooperative effect. When self-assembly of $L^1$ (1.00 equiv) with an equimolar mixture of $Ca(ClO_4)_2 \cdot 4H_2O$ (1.00 equiv) and $La(ClO_4)_3 \cdot 6H_2O$ (1.00 equiv) was performed, absolute metal-selective self-organization, or in other words, narcissistic metal ion self-recognition, was observed, as the [1]H NMR analysis showed only resonances corresponding to $La_4(L^1)_4$ tetrahedral complexes (Supplementary Fig. 140). The exclusive formation of the homometallic $La_4(L^1)_4$ complex was also confirmed by ESI-TOF-MS, in which only multiple charged species ascribed to $[La_4(L^1)_4(ClO_4)_m - nH]^{(12-m-n)+}$ were found (Supplementary Fig. 240).

Analogous to the above self-assembly process, reaction of $L^1$ (1.00 equiv) with metal ion mixtures of $Cd^{II}/La^{III}$ (1.00 equiv of each) led to the formation of homometallic tetrahedra of $La_4(L^1)_4$, as ascertained by [1]H NMR and ESI-TOF-MS (Supplementary Figs. 141 and 241). This absolute self-organization behavior in the mixture of metal ions of identical coordination geometry properties is likely due to the strong supramolecular cooperative mechanical-coupling effect on the tetrahedral cages, facilitating the distinction between metal ions with different electron configurations, ionic charges, and ionic radii.

As for rare earth elements themselves, such absolute self-recognition is much more challenging and intriguing, considering their inherent similar physical and chemical properties, and thus an intricate mixture of heterometallic complexes is predicted in

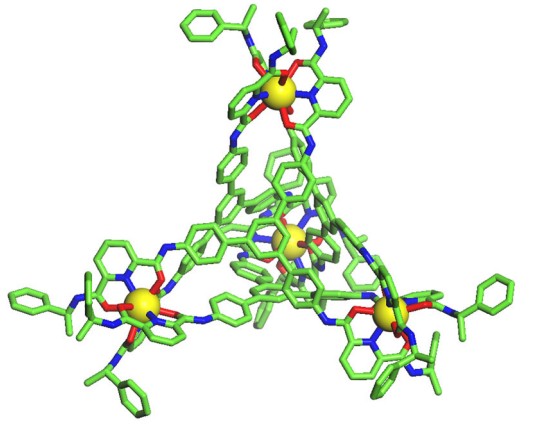

**Fig. 2** X-ray crystal structure of $Cd_4(L^1)_4$. For clarity, only the tetrahedral cage framework is shown. Color code for Cd: Yellow, C: green, N: blue, O: red

the mixed-lanthanide complexation process. To our delight, reaction of $L^1$ (1.00 equiv) with an equimolar mixture of $La^{III}$/$Eu^{III}$ (1.00 equiv of each) turned out to be an absolute self-sorting process, with $Eu_4(L^1)_4$ as the only product, as confirmed by $^1H$

NMR and ESI-TOF-MS (Supplementary Figs. 148 and 247). This complete metal-selective binding phenomenon was totally beyond our expectation as the difference in the ionic radii of $La^{III}$ and $Eu^{III}$ is only 0.10 Å, and in general isostructural compounds will be formed from the same ligand[25].

More surprisingly, highly efficient metallic self-organization was also discovered in the case of lanthanide pairs with much smaller ionic radii difference. For example, reaction of ligand $L^1$ (4 μmol) with an equimolar mixture of $La^{III}$/$Ce^{III}$ (4 μmol of each) resulted in $Ln_4(L^1)_4$ tetrahedral coordination cages containing 10.7 % $La^{III}$ and 89.3% $Ce^{III}$, as determined by $^1H$ NMR spectroscopy (Fig. 3). Two sets of signals were observed in the $^1H$ NMR spectrum of the mixed-metal self-assembled complexes, corresponding to the $La^{III}$- and $Ce^{III}$-coordination environments[27], as the chemical shifts of proton resonances on the ligand are mainly affected by the coordination environment. The proton signals are identified through comparison with homometallic complexes and further confirmed through post-synthetic metathesis experiments (Supplementary Figs. 142, 197, and 242). Based on the highly symmetrical $^1H$ NMR spectrum and the ESI-TOF-MS data, we speculated that $Ce_4(L^1)_4$ and trace amounts of hetero-metallic tetrahedral complex $La_1Ce_3(L^1)_4$ were formed. Nonlinear curve fitting of simulated isotopic patterns in the mass spectrum revealed a composition of 90% $Ce^{III}$ and 10% $La^{III}$, which is in accordance with the $^1H$ NMR analysis (Fig. 3d, Supplementary Figs. 242 and 243). The formation of $Ln_4(L^1)_4$ tetrahedral cages was also confirmed by DOSY spectra (Supplementary Fig. 144). The $^1H$ NMR spectrum of the mixed-metal complexes appears to be insensitive to variations in reaction time and temperature (Supplementary Fig. 143), which indicates that the selectivity is thermodynamically favored.

Given the above observation of effective metal ion self-recognition properties, additional one-pot mixed-metal self-assembly experiments (28 combinations in total) were performed to further investigate the degree of metal ion self-recognition (Supplementary Figs. 145–173 and 244–272). In general, $L^1$ demonstrates a clear and high preference for smaller sized metal ions in multi-component self-assembly process along the lanthanide series and the selectivity increases with the ionic radii difference, making it an ideal candidate for high-efficiency lanthanide separation and purification (Fig. 4, Supplementary Table 3). Moreover, one-pot tri-metallic self-assembly experiments were also conducted and selectivity results comparable to

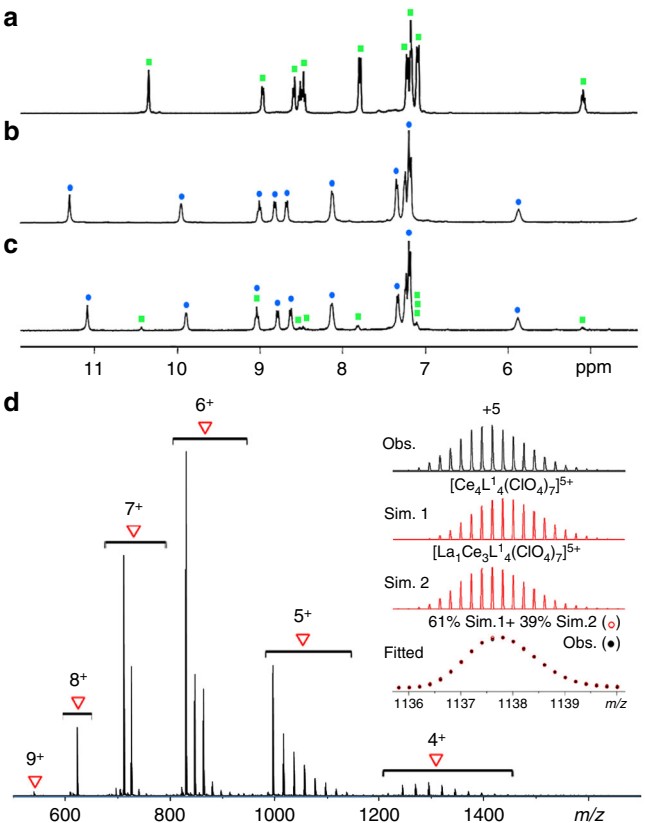

**Fig. 3** Characterization of mixed-metal self-assembled complexes. $^1H$ NMR spectra (400 MHz, $CD_3CN$, 298 K) of **a** $[La_4(L^1)_4](ClO_4)_{12}$, **b** $[Ce_4(L^1)_4]$ $(CF_3SO_3)_{12}$, and **c** $La^{III}$/$Ce^{III}$ mixed-metal self-assembled complexes. ESI-TOF-MS spectrum **d** of the $La^{III}$/$Ce^{III}$ mixed-metal self-assembled complexes ($ClO_4^-$ salts) with insets showing the observed (Obs.) and fitted isotope patterns of the 5 + peak from simulations (Sim.) of the component signals

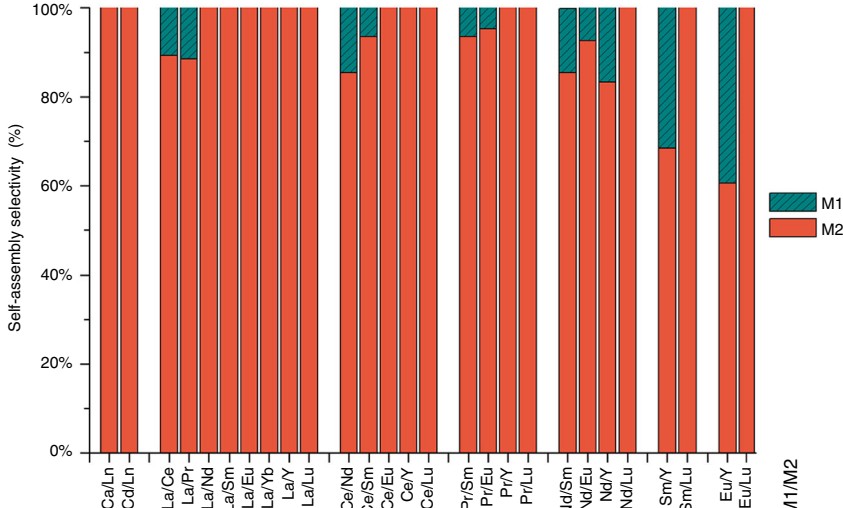

**Fig. 4** Selective self-assembly of $L^1$ with $M_2$ over $M_1$ in the presence of excess equimolar mixture of $M_1$/$M_2$. Self-assembly selectivity was defined as $[M_2]$/$[M_1] + [M_2]$ in the assembled complexes and determined by $^1H$ NMR spectroscopy with ± 5% exp. error

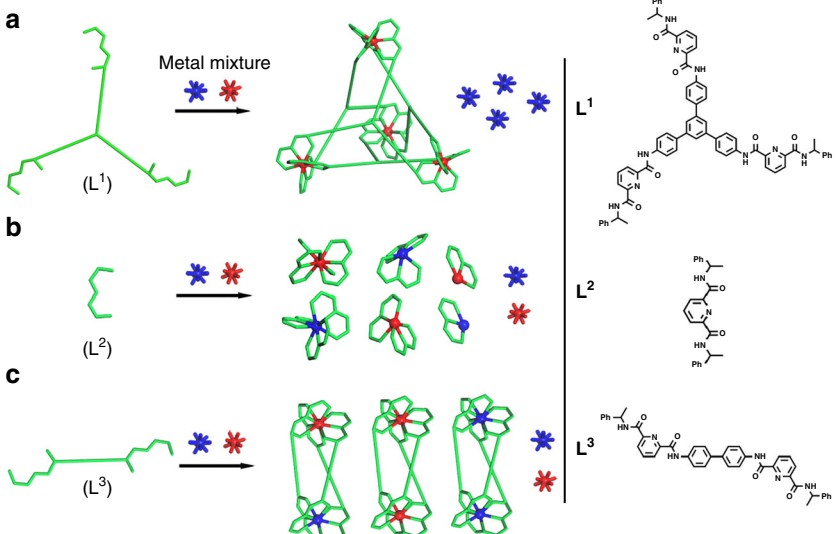

**Fig. 5** Cartoon representations for metal-ion self-recognition. Self-assembly of ligands **a** $L^1$, **b** $L^2$, and **c** $L^3$ with a mixture of alkaline earth ($AE^{II}$), transition ($TM^{II}$) and lanthanide ($Ln^{III}$) metal ions

those of bimetallic systems were obtained, thus expanding the high self-recognition ability of $L^1$ to multi-metal self-assembled systems (Supplementary Figs. 195, 196 and 273, 274). Mixed-metal self-assembly experiments with total metal ions to ligand ratios equal to 1:1 ($Ln^a/Ln^b/L^1 = 0.5/0.5/1$) were also carried out, resulting in the biased formation of two homometallic cages instead of statistically-distributed mixtures of $[Ln^a{}_nLn^b{}_{4-n}(L^1)_4]^{12+}$ ($n = 0$–4) species, especially with lanthanide pair of $La^{III}/Lu^{III}$, implying the powerful metal ion selectivity of $L^1$ (Supplementary Figs. 174–176 and 275–279).

We envisioned that the tiny difference in lanthanide ions was amplified by strong cooperative mechanical-coupling effects during the multi-component self-assembly process, resulting in the observed high fidelity metal ion self-recognition behavior. A series of experiments were conducted to verify the vital function of supramolecular cooperativity in this surprising lanthanide separation and purification.

## Discussion

To verify the extent of the multivalent cooperative effect in the unprecedented selective formation of $M_4L_4$ tetrahedral cages, ligands $L^2$ and $L^3$ (Fig. 5), which contain the same pyridine-2,6-dicarboxamide chelating moiety and are known to form mononuclear $M(L^2)_3$ and dinuclear $M_2(L^3)_3$ complexes with lanthanides, were synthesized according to known procedures[22,60]. Titration experiments were performed in $CD_3CN/CDCl_3$ for $L^{1-3}$ and $Ca^{II}$ with increasing [M]/[L] ratios ($R$) for comparison of structure integrity. The stoichiometry of the tetrahedral assembly $Ca_4(L^1)_4$ was confirmed by [1]H NMR varying $R$ from 0 to 2.0 (Supplementary Fig. 107). It is noteworthy that intermediate spectra ($0.2 \le R_{Ca/L1} \le 1.0$) are simply additions of the ligand and tetrahedral assembly spectra. Moreover, the $Ca_4L^1{}_4$ complexes maintained structural integrity even when the ratio of $Ca^{II}/L^1$ increased to 5.0. This high stability of $M_4(L^1)_4$ toward excess metal ions and ligands serves as a prerequisite for the metal ion self-recognition experiments discussed above.

However, during a similar titration experiment with $L^2$, [1]H NMR, and ESI-TOF-MS showed a mixture of $Ca(L^2)_3$, the observed predominant $Ca^{II}$-containing species, and abundant free ligands at $R < 0.33$ and a mixture of $Ca(L^2)_n$ ($n = 1$–3) at $R > 0.33$ (Supplementary Figs. 110 and 236). This indicates the low stability of $Ca(L^2)_3$ in comparison with $Ca_4(L^1)_4$ tetrahedral cages. In

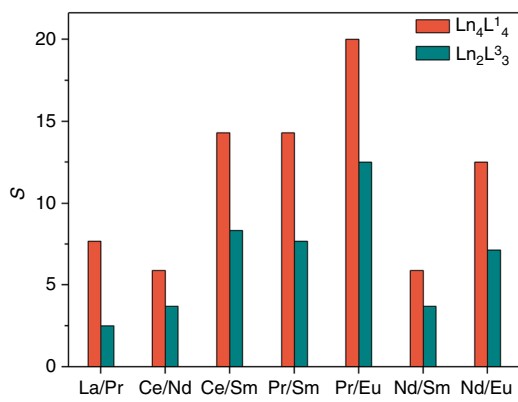

**Fig. 6** Comparison of metal ion selectivity $S$ in $Ln_2L^3{}_3$ and $Ln_4L^1{}_4$ complexes. $S$, defined as $[Ln^b(III)]/[Ln^a(III)]$ and determined by [1]H NMR spectroscopy

a similar titration experiment with $L^3$ ($0.13 \le R_{Ca/L3} \le 2.00$, in $CD_3CN/CDCl_3 = 1/3$), $CaL^3{}_n$ complexes were speculated to form considering the [1]H NMR spectra, nevertheless, large amounts of free ligands were observed in the ESI-TOF-MS, together with small signals from $CaL^3{}_n$ ($n = 1$–3), which can be ascribed to the high fragility of the complexes caused by the low association capacity of $Ca^{II}$-containing complexes (Supplementary Figs. 113 and 238). Titration experiments of $Cd^{II}$ and $Eu^{III}$ with $L^{1-3}$ gave similar variation tendency of stability as that of $Ca^{II}$ (Supplementary Figs. 106–114 and 234–239). The enhanced structural stability from monometallic, dimetallic to tetrametallic self-assembly complexes followed the trend of increasing number of components and coordination interactions, which in turn confirmed the multivalent cooperative effect on the structural integrity of self-assembly systems.

As $M(L^2)_3$ is not applicable to the mixed-metal complexation process due to the rather low structural stability, metal ion selectivity during the formation of $M_2L_3$ and $M_4L_4$ was then compared using ligands $L^3$ and $L^1$ under similar experimental procedure and it was found that the ditopic ligand $L^3$ has much poorer ion selectivity compared with $L^1$ (Fig. 6, Supplementary Figs. 177–188 and 280–288).

The highly efficient recognition observed during the mixed-metal self-assembly process indicates a substantial difference in

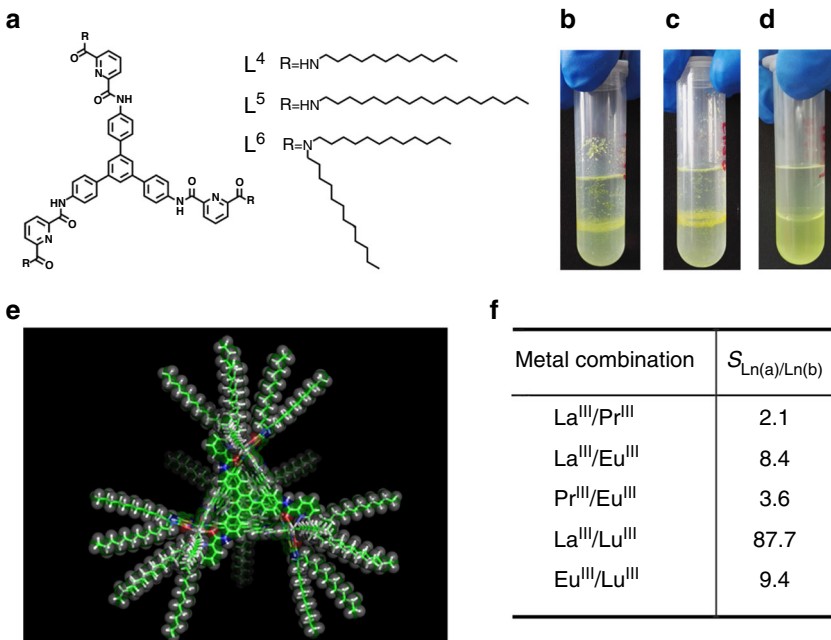

**Fig. 7** Demonstration of a lanthanide separation strategy using the multivalent cooperative effect of the tetranuclear cage complexes. **a** Modified structures of ligands L[4-6] for solvent extraction experiments; Phase separation properties for La[III]/Lu[III] mixed-metal self-assemblies using **b** L[4], **c** L[5], and **d** L[6]; **e** Simulated structure of $Ln_4L^6_4$; **f** Separation factors (with ± 5% exp. error) obtained from lanthanide solvent extraction experiment

| Metal combination | $S_{Ln(a)/Ln(b)}$ |
|---|---|
| La[III]/Pr[III] | 2.1 |
| La[III]/Eu[III] | 8.4 |
| Pr[III]/Eu[III] | 3.6 |
| La[III]/Lu[III] | 87.7 |
| Eu[III]/Lu[III] | 9.4 |

the binding affinity of L[1] toward different lanthanide ions[61-63]. Post-synthetic metal-metathesis experiments were conducted to shed light on the mechanism of this highly controlled metal-selective self-assembly process. It is worth mentioning that the substitution rate and idealized relative formation constants for each metal combination depend on the difference in the ionic radii, and a larger difference results in faster substitution process and larger relative formation constants (Supplementary Figs. 197–203 and 295). $Eu_4(L^1)_4$, for instance, has a formation constant of at least nine orders of magnitude higher than that of $La_4(L^1)_4$ (Supplementary Table 4). Such a huge difference in formation constants serves as the primary driving force for the highly efficient metal ion recognition, and explains the complete Eu[III] selectivity observed in the La[III]/Eu[III] mixed-metal self-assembly discussed above.

Post-synthetic metal-metathesis experiments were also performed using L[3] for comparing relative formation constants with L[1] (Supplementary Figs. 204–208). Compared with the $Ln_4(L^1)_4$ complexes, $Ln_2(L^3)_3$ has much smaller relative formation constants for the same metal combinations (Supplementary Table 5). The increased thermodynamic stability and metal ion selectivity on going from dimetallic to tetrametallic complexes confirm that the cooperativity is dramatically enhanced with increased numbers of multitopic ligand chelation.

In addition to supramolecular multivalent cooperativity, structural stability of the supramolecular polyhedra and the rational choice of the multidentate coordination sites also contribute to the high-efficient metal ion selectivity. As a precondition for efficient metal ion selective self-assembly, the structural stability relies on both framework rigidity of the ligand and chelating affinity of the coordinating moieties for metal ions. Hamacek's group has reported a pyridine-2,6-dicarboxamide (pcam) based tripodal ligand with flexible bridging units 1,1,1-tris(aminomethyl)ethane[64]. Tetrahedral complexes $[Ln_4L_4]^{12+}$ (Ln = Eu, Tb, Lu) were formed in the self-assembly process. [1]H NMR and ESI-MS titration show the appearance of other species when either ligands ($LnL_3$, $LnL_2$, $Ln_2L_3$, $Ln_3L_4$) or metal ions ($Ln_4L_3$,

$Ln_3L_2$) are in excess. Moreover, the [1]H NMR spectrum excluded the formation of a tetrahedral complex with La[III]. The rather low stability is speculated to derive from the flexibility of the ligand. Moreover, the $Ln_4L_4$ complexes assembled from the flexible ligand have rather small relative formation constants, with $log\beta_{La/Eu} = 0.5$ and $log\beta_{Tb/Lu} = 1.4$ (ref. [61]). Reaction of lanthanide ions with a rigid tripodal ligand, three pcam coordination units connected with rigid triptycene moiety, generates low symmetry complexes in presence of excess ligand and a trinuclear sandwich complex $[Eu_3L_2]^{9+}$ when the metal/ligand ratio [Eu]/[L] reaches 3:2 (ref. [65]). For tris(tridentate) ligands with similar coordination moieties, both scaffold rigidity and geometry exert great influence on the structural stability. We hypothesize ligands with moderate rigidity operate as levers between the four metal centers on the tetrahedral vertices, in a way that a small distortion in coordination geometry on one metal center is transferred to the other three vertices, leading to an enlarged energy barrier in comparison with the perfectly symmetrical tetrahedral framework. However, changing one of the metal ions on a flexible tetrahedron does not have such an effect. Thus, we conclude that ligand rigidity enhances mechanical-coupling effects within the framework, and in turn contributes to cooperative enhancement of both stability and metal ion selectivity.

Higher chelating affinity between the coordinating moieties and metal ions would improve the structural stability but it has a complicated impact on metal ion selectivity. Ligands with ideal coordination sites for lanthanide separation in a supramolecular system are expected to coordinate with the entire lanthanide series while possessing distinct binding affinity toward different lanthanide ions. However, in addition to the efficient chelating ability of the ligand with all the lanthanide ions into specific structures, moderate binding affinity is necessary for practical usage. Overlarge binding affinity could result in kinetically trapped assemblies and hinder further transformation into the thermodynamically favored product (needing high temperature or prolonged reaction time). Hooley's group has reported an acylhydrazone-phenolate based bis(tridentate) ligand with

anionic coordination sites, which assembles with lanthanide ions into $Ln_2L_3$ complexes in DMSO and shows kinetic discrimination among lanthanide ions with a preference for smaller metals and a thermodynamic preference for larger metals[25]. However, in this system, the thermodynamic equilibrium is not reached even after 20 h. In comparison, the pcam-based $Ln_4(L^1)_4$ complexes dissociate in DMSO, implying much weaker binding affinity between ligands and metal ions and the thermodynamic equilibrium is reached on a minute timescale. Supramolecular systems that can rapidly achieve thermodynamic equilibrium are required for efficient lanthanide separation for economic and practical consideration. Moreover, the pcam-based ditopic and tritopic ligands possess much higher lanthanide ion selectivity in the thermodynamically favored complexes owing to their moderate chelating affinity (Supplementary Figs. 186–188 and Table 2).

Cooperative enhancement of lanthanide selectivity in the formation of the tetrahedral cages indicated that tris(tridentate) ligands may serve as good extractants for lanthanide separation. As a proof-of-concept, $L^{4–6}$, with hydrophobic alkyl groups introduced onto the periphery to afford better phase separation (Fig. 7), were synthesized and tested in lanthanide extraction experiments. The introduction of di-dodecanamine groups ($L^6$) finally gave good dispersity of the complexes in $CHCl_3$ for liquid–liquid extraction (Fig. 7b–d). In a typical procedure, $L^6$ (12 μmol) was treated with an equimolar mixture of $La(OTf)_3$ and $Lu(OTf)_3$ (12 μmol of each) in 2 mL mixed solvent of $CH_3CN$/ $CHCl_3$ (1:1 v/v, for better solubility) at room temperature. The turbid suspension of ligands became clear within 5 min with gentle shaking, and [1]H NMR and ESI-TOF-MS confirmed the exclusive formation of $Lu_4(L^6)_4$ complexes (Supplementary Figs. 192 and 292). After the reaction solvent was evaporated under reduced pressure, 2 mL $CHCl_3$ was added to the self-assembled complex system, followed by the addition of the same volume of water to extract the unreacted $La(OTf)_3$. The structural integrity of $Lu_4(L^6)_4$ in the organic phase after extraction was ascertained by [1]H NMR spectroscopy and the metal contents in the two separated phases were measured using inductively coupled plasma mass spectrometry (ICP-MS). The separation factor, defined as the ratio of the distribution coefficient of each lanthanide in the aqueous and organic phase ($S_{Ln(a)/Ln(b)} = D_{Ln(a)}/D_{Ln}$ $(b)$, distribution coefficient $D_{Ln(a)} = [Ln_{(a)}]_{aq}/[Ln_{(a)}]_{org}$), was calculated to be ca. 87.7 without further optimization of the extraction process (with ± 5% exp. error). Further separation factors measured for some representative metal combinations are listed in Fig. 7f (Supplementary Figs. 189–194 and 289–294). Parallel extraction experiments were also conducted, suggesting good validity of the separation efficiency (Supplementary Table 6). In view of the poor water stability of the core cage compound, which in fact will fall apart when exposed to $CD_3CN/D_2O$ (1:1 v/v) mixed solvent, we anticipate that separation factors can be further increased with this strategy by employing more stable tetrahedral frameworks. Furthermore, this supramolecular separation strategy is very promising in efficient actinides/actinides and actinides/lanthanides separation for the treatment of radioactive waste and the recycling of minor actinides, considering the similarities in oxidation states, chemical properties and ionic radii between actinides and lanthanides.

In summary, this supramolecular lanthanide extraction and separation approach has been established based on an exclusive metal ion self-sorting of tetrahedral cage complexes. Unprecedented separation abilities have been achieved by taking advantage of the multivalent supramolecular cooperativity of these complexes. As such, this study provides new insights into the design of next-generation lanthanide extractants. Further application of this strategy to lanthanide/actinide separation, or purification of other metals in general, is expected.

## Methods

**Materials**. Deuterated solvents were purchased from Admas and J&K scientific. 1D and 2D NMR spectra were measured on a Bruker-BioSpin AVANCE III HD (400 MHz) spectrometer. [1]H NMR chemical shifts were determined with respect to residual solvent signals of the deuterated solvents used. [1]H NMR integrations were performed using TOPSPIN 2.1 software. ESI-TOF-MS were recorded on an Impact II UHR-TOF mass spectrometry from Bruker, with ESI-L low concentration tuning mix (from Agilent Technologies) as the internal standard (Accuracy < 3 ppm). Data analyses and simulations of ESI-TOF-MS were processed with the Bruker Data Analysis software (Version 4.3). ICP-MS analysis was performed on a Thermo Finnigan high-resolution magnetic sector Element 2 ICP-MS instrument. Unless otherwise stated, all chemicals and solvents were purchased from commercial companies and used without further purification.

Caution! Perchlorate salts are potentially explosive and should be handled carefully in small quantities.

**Synthesis and physical properties of $[M_4L^1_4]^{8+}$ and $[Ln_4L^1_4]^{12+}$**. A solution of $M(CF_3SO_3)_2$ (or $M(ClO_4)_2•4H_2O$, M = Ca, Cd) or $Ln(ClO_4)_3•6H_2O$ (10.0 μmol, 1 equiv) (Ln = La, Pr, Nd, Sm, Eu, Yb, Y) in 0.50 mL $CH_3CN$ was added to a white suspension of $L^1$ (either the $R$ or $S$ enantiomer form; 11.08 mg, 10.0 μmol, 1 equiv) in 1.00 mL $CH_3CN$. After stirring at 40 °C for 1 h, the turbid suspension of ligands turned into homogeneous yellow solution. [1]H NMR and ESI-TOF-MS showed the quantitative formation of $M_4(L^1)_4$ complexes. The solvent is removed under reduced pressure to give a yellow powder product (Supplementary Figs. 29–72 and 209–219).

The above experimental procedure applies to the self-assembly of $Ln(CF_3SO_3)_3$ (Ln = La, Ce, Pr, Nd, Sm, Eu, Yb, Lu, Y) with $L^1$ as well. And it is worth mentioning that $Ce(ClO_4)_3•6H_2O$ and $Lu(ClO_4)_3•6H_2O$ are not used in the preparation of $[Ln_4(L^1)_4]^{12+}$ due to the poor solubility of their self-assembled complexes in $CH_3CN$. However, in the existence of extra Ln(III) in mixed-metal one-pot self-assembly experiments, $[Ce_4(L^1)_4](ClO_4)_{12}$ and $[Lu_4(L^1)_4](ClO_4)_{12}$ have rather good solubility in $CH_3CN$, which facilitates the manipulation and characterization of the metal-selective self-assembly process. No change in the NMR spectra was observed for the $Ln_4(L^1)_4$ complexes with either $ClO_4^-$ or $CF_3SO_3^-$ as the counter anions. No signals were observed in the negative range (−20–0 ppm) in the [1]H NMR spectra.

$[Ca_4L^1_4](CF_3SO_3)_8$: [1]H NMR (400 MHz, $CD_3CN$) δ 9.90 (s, 3 H), 8.44 (d, $J$ = 7.2 Hz, 3 H), 8.38–8.23 (m, 9 H), 7.88 (d, $J$ = 8.5 Hz, 6 H), 7.31 – 7.03 (m, 24 H), 5.16–5.03 (m, 3 H), 1.71 (d, $J$ = 7.1 Hz, 9 H). [13]C NMR (101 MHz, $CD_3CN$) δ 165.11, 164.55, 149.26, 148.93, 143.54, 141.26, 140.16, 137.01, 128.98, 127.82, 127.19, 126.65, 125.07, 124.73, 123.14, 122.91, 122.63, 122.04, 119.96, 51.41, 21.38. ESI-TOF-MS calcd. for $[M-7(CF_3SO_3^-)]^{7+}$ 677.3686, found 677.3736; calcd. for $[M-6(CF_3SO_3^-)]^{6+}$ 815.0887, found 815.0914; calcd. for $[M-5(CF_3SO_3^-)]^{5+}$ 1007.8970, found 1007.9010; calcd. for $[M-4(CF_3SO_3^-)]^{4+}$ 1297.1093, found 1297.3668.

$[Cd_4L^1_4](ClO_4)_8$: [1]H NMR (400 MHz, $CD_3CN$) δ 9.80 (s, 3 H), 8.49–8.32 (m, 12 H), 7.83 (d, $J$ = 8.6 Hz, 6 H), 7.35–7.12 (m, 24 H), 5.12 (p, $J$ = 7.1 Hz, 3 H), 1.74 (d, $J$ = 7.0 Hz, 9 H). [13]C NMR (101 MHz, $CD_3CN$) δ 163.61, 162.89, 146.76, 146.45, 143.65, 141.69, 140.23, 137.09, 136.96, 129.05, 127.88, 127.08, 126.71, 125.70, 125.46, 122.82, 122.67, 51.66, 21.51. ESI-TOF-MS calcd. for $[M-8(ClO_4^-)]^{8+}$ 610.2990, found 610.6775; calcd. for $[M-7(ClO_4^-)]^{7+}$ 711.7629, found 711.6234; calcd. for $[M-6(ClO_4^-)]^{6+}$ 846.8814, found 846.8852; calcd. for $[M-5(ClO_4^-)]^{5+}$ 1036.2475, found 1036.2516; calcd. for $[M-4(ClO_4^-)]^{4+}$ 1320.0464, found 1320.3012.

$[La_4L^1_4](ClO_4)_{12}$: [1]H NMR (400 MHz, $CD_3CN$) δ 10.35 (s, 3 H), 8.96 (d, $J$ = 6.4 Hz, 3 H), 8.59 (d, $J$ = 7.6 Hz, 3 H), 8.52 (d, $J$ = 8.0 Hz, 3 H), 8.47 (t, $J$ = 7.8 Hz, 3 H), 7.79 (d, $J$ = 8.4 Hz, 6 H), 7.22 (d, $J$ = 8.0 Hz, 6 H), 7.17 (s, 9 H), 7.09 (t, $J$ = 5.2 Hz, 9 H), 5.09 (t, $J$ = 6.8 Hz, 3 H), 1.70 (d, $J$ = 8.0 Hz, 9 H). [13]C NMR (101 MHz, $CD_3CN$) δ 168.17, 167.35, 149.43, 149.09, 143.85, 142.34, 140.34, 138.56, 135.62, 129.41, 129.11, 128.18, 127.67, 127.27, 126.39, 123.91, 123.57, 117.98, 52.78, 29.91, 21.27. ESI-TOF-MS calcd. for $[M-9(ClO_4^-)-3(HClO_4)]^{9+}$ 553.9310, found 553.9323; calcd. for $[M-8(ClO_4^-)-4(HClO_4)]^{8+}$ 623.0465, found 623.0479; calcd. for $[M-7(ClO_4^-)-3(HClO_4)]^{7+}$ 740.6109, found 740.6123; calcd. for $[M-6(ClO_4^-)-4(HClO_4)]^{6+}$ 863.8781, found 863.8795; calcd. for $[M-5(ClO_4^-)-4(HClO_4)]^{5+}$ 1056.4434, found 1056.4447; calcd. for $[M-4(ClO_4^-)-4(HClO_4)]^{4+}$ 1345.5414, found 1345.5422.

$[Ce_4L^1_4](CF_3SO_3)_{12}$: [1]H NMR (400 MHz, $CD_3CN$) δ 11.38 (s, 3 H), 10.07 (d, $J$ = 4.4 Hz, 3 H), 8.99 (t, $J$ = 8.0 Hz, 3 H), 8.84 (d, $J$ = 8.0 Hz, 3 H), 8.72 (d, $J$ = 8.0 Hz, 3 H), 8.12 (d, $J$ = 7.2 Hz, 6 H), 7.35 (d, $J$ = 7.2 Hz, 6 H), 7.23 (d, $J$ = 5.2 Hz, 6 H), 7.16 (t, $J$ = 5.2 Hz, 12 H), 5.83 (s, 3 H), 1.94 (d, $J$ = 7.2 Hz, 3 H). [13]C NMR (101 MHz, $CD_3CN$) δ 164.99, 164.03, 146.87, 146.53, 142.72, 142.54, 140.47, 138.65, 136.13, 131.63, 131.40, 129.14, 128.13, 127.88, 126.56, 123.96, 123.71, 123.04, 119.86, 53.65, 21.47. ESI-TOF-MS calcd. for $[M-8(CF_3SO_3^-)-4(HCF_3SO_3)]^{8+}$ 623.5460, found 623.5462; calcd. for $[M-7(CF_3SO_3^-)-5(HCF_3SO_3)]^{7+}$ 712.4800, found 712.4802; calcd. for $[M-6(CF_3SO_3^-)-6(HCF_3SO_3)]^{6+}$ 831.0588, found 831.2261; calcd. for $[M-5(CF_3SO_3^-)-7(HCF_3SO_3)]^{5+}$ 997.0692, found 997.2694; calcd. for $[M-4(CF_3SO_3^-)-6(HCF_3SO_3)]^{4+}$ 1321.3150, found 1321.3137.

$[Eu_4L^1_4](ClO_4)_{12}$: [1]H NMR (400 MHz, $CD_3CN$) [1]H NMR (400 MHz, $CD_3CN$) δ 8.67 (s, 6 H), 7.94 (d, $J$ = 7.3 Hz, 6 H), 7.73 (s, 3 H), 7.22 (d, $J$ = 7.8 Hz, 3 H), 7.05 (s, 9 H), 6.94 (d, $J$ = 19.8 Hz, 9 H), 6.45 (d, $J$ = 7.2 Hz, 3 H), 6.38 (d, $J$ = 7.8 Hz, 3 H), 5.95 (s, 3 H), 4.76 (s, 3 H), 1.93 (d, $J$ = 5.7 Hz, 9 H). [13]C NMR (101 MHz, $CD_3CN$) δ

164.746, 159.907, 156.081, 143.316, 142.099, 140.586, 139.522, 135.987, 129.004, 127.951, 127.855, 125.991, 125.174, 124.671, 117.910, 92.685, 92.290, 52.082, 22.246. ESI-TOF-MS calcd. for $[M-8(ClO_4^-)-4(HClO_4)]^{8+}$ 629.5535, found 629.5544; calcd. for $[M-7(ClO_4^-)-5(HClO_4)]^{7+}$ 719.3459, found 719.3468; calcd. for $[M-6(ClO_4^-)-6(HClO_4)]^{6+}$ 839.0690, found 839.0700; calcd. for $[M-5(ClO_4^-)-7(HClO_4)]^{5+}$ 1006.6813, found 1006.6819; calcd. for $[M-4(ClO_4^-)-6(HClO_4)]^{4+}$ 1308.3277, found 1308.5779.

NMR and ESI-TOF-MS characterization of $[Pr_4L^1_4](ClO_4)_{12}$, $[Nd_4L^1_4](ClO_4)_{12}$, $[Sm_4L^1_4](ClO_4)_{12}$, $[Y_4L^1_4](ClO_4)_{12}$, $[Yb_4L^1_4](CF_3SO_3)_{12}$ and $[Lu_4L^1_4](CF_3SO_3)_{12}$ can be seen in Supplementary Methods.

**Synthesis and physical properties of $[Ln_4L^{4-6}_4]^{12+}$.** A solution of $Ln(CF_3SO_3)_3$ (10.0 μmol, 1 equiv) (Ln = La, Ce, Pr, Eu, Lu) in 0.50 mL $CD_3CN$ was added to a suspension of $L^{4-6}$ (10.0 μmol, 1 equiv) in 1.00 mL $CD_3CN/CDCl_3$ (1/1 v/v). Homogeneous yellow solution was obtained after stirring at room temperature for 1 h. NMR and ESI-TOF-MS spectra showed the quantitative formation of $[Ln_4L^{4-6}_4](CF_3SO_3)_{12}$ (Supplementary Figs. 90–105 and 227–233).

$[La_4L^6_4](CF_3SO_3)_{12}$: $^1$H NMR (400 MHz, $CD_3CN$) δ 11.05 (s, 3 H), 8.79 (d, J = 8.1 Hz, 3 H), 8.48 (t, J = 7.9 Hz, 3 H), 8.08 (d, J = 7.8 Hz, 3 H), 7.87 (d, J = 8.4 Hz, 6 H), 7.29 (d, J = 8.4 Hz, 6 H), 7.18 (s, 3 H), 3.63 (s, 3 H), 3.54 (d, J = 6.0 Hz, 3 H), 3.27 (s, 3 H), 3.16 (d, J = 17.8 Hz, 3 H), 1.80 (s, 6 H), 1.46 (s, 6 H), 1.25 (d, J = 16.2 Hz, 96 H), 1.19 (s, 12 H), 0.85 (dd, J = 9.6, 6.9 Hz, 18 H). $^{13}$C NMR (101 MHz, $CD_3CN$) δ 169.29, 167.49, 150.09, 149.62, 143.26, 140.22, 138.63, 135.50, 128.38, 127.42, 123.29, 122.55, 119.37, 51.11, 49.53, 32.17, 32.07, 30.05, 30.01, 29.96, 29.81, 29.76, 29.65, 29.59, 29.51, 29.30, 28.75, 27.38, 26.86, 26.50, 22.89, 22.81, 14.00, 13.95. ESI-TOF-MS calcd. for $[M-8(CF_3SO_3^-)-4(HCF_3SO_3)]^{8+}$ 971.7668, found 971.7673; calcd. for $[M-7(CF_3SO_3^-)-5(HCF_3SO_3)]^{7+}$ 1110.4468, found 1110.4469; calcd. for $[M-6(CF_3SO_3^-)-6(HCF_3SO_3)]^{6+}$ 1295.3533, found 1295.3525; calcd. for $[M-5(CF_3SO_3^-)-7(HCF_3SO_3)]^{5+}$ 1554.2226, found 1554.2221.

$[Pr_4L^6_4](CF_3SO_3)_{12}$: $^1$H NMR (400 MHz, $CD_3CN$) δ 12.56 (s, 3 H), 10.30 (s, 3 H), 10.06 (s, 3 H), 9.40 (s, 3 H), 6.10 (s, 9 H), 5.90 (s, 6 H), 3.68 (d, J = 24.7 Hz, 6 H), 2.96 (s, 3 H), 2.55 (s, 3 H), 2.07 (s, 3 H), 1.87 (s, 3 H), 1.24 (d, J = 11.3 Hz, 96 H), 0.91 (d, J = 5.8 Hz, 6 H), 0.90–0.83 (m, 18 H), 0.70 (s, 6 H), 0.54 (s, 3 H). ESI-TOF-MS calcd. for $[M-8(CF_3SO_3^-)-4(HCF_3SO_3)]^{8+}$ 972.7675, found 972.7676; calcd. for $[M-7(CF_3SO_3^-)-5(HCF_3SO_3)]^{7+}$ 1111.5904, found 1111.5908; calcd. for $[M-6(CF_3SO_3^-)-6(HCF_3SO_3)]^{6+}$ 1296.6875, found 1296.6874; calcd. for $[M-5(CF_3SO_3^-)-7(HCF_3SO_3)]^{5+}$ 1555.8236, found 1555.8223; calcd. for $[M-4(CF_3SO_3^-)-6(HCF_3SO_3)]^{4+}$ 2019.5074, found 2019.5039.

$[Eu_4L^6_4](CF_3SO_3)_{12}$: $^1$H NMR (400 MHz, $CD_3CN$) δ 8.46 (s, 6 H), 7.84 (s, 6 H), 7.65 (s, 6 H), 7.29 (s, 3 H), 6.97 (s, 3 H), 6.23 (s, 3 H), 6.23 (s, 1 H), 3.51 (s, 6 H), 3.40 (s, 3 H), 3.13 (s, 3 H), 1.73 (s, 6 H), 1.21 (t, J = 47.2 Hz, 114 H), 1.02–0.64 (m, 18 H). $^{13}$C NMR (101 MHz, $CD_3CN$) δ 154.77, 145.93, 140.23, 139.37, 135.52, 127.36, 124.68, 95.36, 94.25, 49.28, 47.79, 32.11, 32.00, 30.09, 30.04, 29.97, 29.93, 29.75, 29.70, 29.64, 29.60, 29.44, 29.27, 27.78, 27.53, 26.23, 22.82, 22.74, 13.98, 13.92. ESI-TOF-MS calcd. for $[M-8(CF_3SO_3^-)-4(HCF_3SO_3)]^{8+}$ 978.2739, found 978.2744; calcd. for $[M-7(CF_3SO_3^-)-5(HCF_3SO_3)]^{7+}$ 1117.8834, found 1117.8844; calcd. for $[M-6(CF_3SO_3^-)-6(HCF_3SO_3)]^{6+}$ 1304.0294, found 1304.1962; calcd. for $[M-5(CF_3SO_3^-)-7(HCF_3SO_3)]^{5+}$ 1564.6339, found 1564.6327; calcd. for $[M-4(CF_3SO_3^-)-8(HCF_3SO_3)]^{4+}$ 1955.5405, found 1955.5380.

$[Lu_4L^6_4](CF_3SO_3)_{12}$: $^1$H NMR (400 MHz, $CD_3CN$) δ 11.05 (s, 3 H), 8.87 (d, J = 7.6 Hz, 3 H), 8.52 (t, J = 7.6 Hz, 3 H), 8.12 (d, J = 7.6 Hz, 3 H), 7.82 (d, J = 8.0 Hz, 6 H), 7.31 (d, J = 8.0 Hz, 6 H), 7.24 (s, 3 H), 3.72 (s, 3 H), 3.55 (dd, J = 16.9, 12.9 Hz, 3 H), 3.15 (s, 3 H), 1.85 (s, 6 H), 1.40–1.27 (m, 102 H), 1.13 (d, J = 6.9 Hz, 6 H), 1.03 (s, 6 H), 0.90–0.85 (m, 18 H). ESI-TOF-MS calcd. for $[M-8(CF_3SO_3^-)-4(HCF_3SO_3)]^{8+}$ 989.7840, found 989.7853; calcd. for $[M-7(CF_3SO_3^-)-5(HCF_3SO_3)]^{7+}$ 1131.0378, found 1131.0398; calcd. for $[M-6(CF_3SO_3^-)-6(HCF_3SO_3)]^{6+}$ 1319.3763, found 1319.3773; calcd. for $[M-5(CF_3SO_3^-)-7(HCF_3SO_3)]^{5+}$ 1583.0501, found 1583.0501; calcd. for $[M-4(CF_3SO_3^-)-6(HCF_3SO_3)]^{4+}$ 2053.5405, found 2503.7903.

$[Eu_4L^4_4](CF_3SO_3)_{12}$: $^1$H NMR (400 MHz, $CD_3CN$) δ 8.68 (s, 6 H), 7.93 (d, J = 7.3 Hz, 6 H), 7.77 (s, 3 H), 7.49 (s, 3 H), 7.26 (t, J = 8.2 Hz, 3 H), 6.69 (d, J = 7.5 Hz, 3 H), 6.48 (d, J = 7.0 Hz, 3 H), 4.80 (s, 3 H), 3.80 (s, 6 H), 1.77 (s, 6 H), 1.61 (d, J = 11.9 Hz, 6 H), 1.47 (s, 6 H), 1.32 (dd, J = 18.7, 13.8 Hz, 42 H), 0.89 (dd, J = 8.9, 4.4 Hz, 9 H). ESI-TOF-MS calcd. for $[M-9(CF_3SO_3^-)-3(HCF_3SO_3)]^{9+}$ 645.3267, found 645.3273; calcd. for $[M-8(CF_3SO_3^-)-4(HCF_3SO_3)]^{8+}$ 725.8667, found 725.8678; calcd. for $[M-7(CF_3SO_3^-)-4(HCF_3SO_3)]^{7+}$ 850.8408, found 850.8414; calcd. for $[M-6(CF_3SO_3^-)-4(HCF_3SO_3)]^{6+}$ 1017.4730, found 1017.4729; calcd. for $[M-5(CF_3SO_3^-)-4(HCF_3SO_3)]^{5+}$ 1250.7580, found 1250.7570; calcd. for $[M-4(CF_3SO_3^-)-4(HCF_3SO_3)]^{4+}$ 1600.6856, found 1600.6835.

$[Eu_4L^5_4](CF_3SO_3)_{12}$: $^1$H NMR (400 MHz, $CD_3CN$) δ 8.62 (s, 6 H), 7.90 (s, 6 H), 7.73 (s, 3 H), 7.59 (s, 3 H), 7.25 (s, 3 H), 6.78 (s, 3 H), 6.56 (s, 3 H), 4.93 (s, 3 H), 3.78 (s, 6 H), 1.27 (s, 96 H), 0.87 (s, 9 H). ESI-TOF-MS calcd. for $[M-8(CF_3SO_3^-)-4(HCF_3SO_3)]^{8+}$ 852.0077, found 852.0065; calcd. for $[M-7(CF_3SO_3^-)-4(HCF_3SO_3)]^{7+}$ 995.1451, found 995.2864; calcd. for $[M-6(CF_3SO_3^-)-4(HCF_3SO_3)]^{6+}$ 1185.8280, found 1185.8255; calcd. for $[M-5(CF_3SO_3^-)-6(HCF_3SO_3)]^{5+}$ 1392.8002, found 1392.7972; calcd. for $[M-4(CF_3SO_3^-)-6(HCF_3SO_3)]^{4+}$ 1778.2383, found 1778.2319.

Synthesis and characterization of $L^{4-6}$ can be seen in Supplementary Methods and Supplementary Figs. 11–28.

Self-assembly of $L^2$ and $L^3$ with lanthanide ions is carried out according to literature [26,66].

NMR and ESI-TOF-MS characterization of $[Eu_1L^2_2](CF_3SO_3)_3$, $[La_2L^3_3](ClO_4)_6$, $[Ce_2L^3_3](CF_3SO_3)_6$, $[Pr_2L^3_3](ClO_4)_6$, $[Nd_2L^3_3](ClO_4)_6$, $[Sm_2L^3_3](ClO_4)_6$, $[Eu_2L^3_3](ClO_4)_6$ and $[Pr_4L^6_4](CF_3SO_3)_{12}$, can be seen in Supplementary Methods and Supplementary Figs. 73–89, 109–114, 220–226.

**General procedure for mixed-metal one-pot self-assembly of $L^{1,3,6}$.** $L^1$ (6.0 μmol) was treated with an equimolar mixture of $Ln^a(ClO_4)_3 \cdot 6H_2O$ and $Ln^b(ClO_4)_3 \cdot 6H_2O$ (6.0 μmol of each) in $CD_3CN$ (0.6 mL) at 40 °C for 1 h and the turbid suspension of ligands gradually turned clear. The resulting yellow solution was characterized by $^1$H NMR and ESI-TOF-MS to identify the selectivity in the mixed-metal one-pot self-assembly process. No change in the $^1$H NMR spectra was observed after elongated reaction time for even 2 weeks, suggesting that the self-assembly is fast and thermodynamically stable (Supplementary Figs. 116–176, 240–279).

$^1$H NMR spectra of the mixed-metal self-assembled complexes with non-absolute self-recognition behavior were measured with d1 value set as 20 s to ensure the accuracy of the metal ion selectivity, which was calculated based on $^1$H NMR integration.

Metal combinations with rather small ionic difference were avoided, such as $Sm^{III}/Eu^{III}$, which leads to the formation of an intricate mixture of complexes with low symmetry as a result of poor metal-ion selectivity, making it difficult for the identification of selectivity through $^1$H NMR integration.

The selectivity for smaller sized $Ln^{III}$ in incomplete self-sorting mixed-metal self-assembly process is determined by $^1$H NMR. The highly symmetrical $^1$H NMR patterns excluded the formation of a dynamic mixture of scrambled-metal cages. ESI-TOF-MS analyses further confirmed the formation of trace amounts of mono substituted $(Ln_1^aLn^b_3L^1_4)$ cage, along with homometallic cages. As the chemical shifts of the signals arising from the tetrahedral assemblies are manifestation of magnetic environments imposed by the coordinated paramagnetic $Ln^{III}$, integration of two sets of NMR signals can be used for quantification of two kinds of $Ln^{III}$ vertices (Supplementary Figs. 242–274).

$Cd^{II}/Ln^{III}$ or $Ca^{II}/Ln^{III}$ mixed-metal one-pot self-assembly with $L^1$ was implemented in a similar procedure as above. As ligand $L^1$ has much higher self-assembly preference to $Ln^{III}$ ions with smaller ionic radii along the lanthanide series, mixed-metal self-assembly of $Cd^{II}/Ln^{III}$ and $Ca^{II}/Ln^{III}$ were only proceeded for $La^{III}$, which has smaller association constant than other lanthanide ions and complete mixed-metal self-sorting assembly of $Cd^{II}/Ln^{III}$ and $Ca^{II}/Ln^{III}$ was thus speculated (Supplementary Figs. 140–141 and 240–241).

**Nonlinear curve fitting of simulated isotope patterns of $La^{III}$-$Ce^{III}$ mixed-metal self-assembly complexes with $L^1$.** Considering the tiny difference in the ESI response factors of the tetrahedral complexes of different lanthanide ions, nonlinear curve fitting using the model of $x[Ce_4L^1_4]^{12+}$ and $(1-x)[La_1Ce_3L^1_4]^{12+}$ fits well with the observed mass spectrum, with the composition of $[Ce_4L^1_4]^{12+}/[La_1Ce_3L^1_4]^{12+}$ in the $La^{III}$-$Ce^{III}$ mixed-metal self-assembly complexes as 0.61/0.39 and 0.62/0.38 for 5 + and 6 + peaks, respectively. This means about ten percent of $La^{III}$ is incorporated in the mixed-metal complexes, which agrees well with $^1$H NMR analysis of 10.7 percent $La^{III}$. Similar fitting using the model of $x[Ce_4L^1_4]^{12+}$ and $(1-x)[La_4L^1_4]^{12+}$ did not give consistent results with the observed isotope patterns of the 5 + and 6 + peaks (Supplementary Figs. 242–243).

**General procedure for mixed-metal one-pot self-assembly of $L^3$.** $L^{3R}$ or $L^{3S}$ (4.5 μmol) was treated with an equimolar mixture of $Ln^a(ClO_4)_3 \cdot 6H_2O$ and $Ln^b(ClO_4)_3 \cdot 6H_2O$ (3.0 μmol of each) in $CD_3CN$ (0.6 mL) at 40 °C for 1 h and the turbid suspension of ligands gradually turned clear. The resulting yellow solution was characterized by $^1$H NMR and ESI-TOF-MS to identify the selectivity in the mixed-metal one-pot self-assembly process. No change in the $^1$H NMR spectra was observed after elongated reaction time for even 1 month, suggesting the self-assembly is fast and thermodynamically stable (Supplementary Figs. 177–188 and 280–288).

**General procedure for mixed-metal one-pot self-assembly of $L^6$.** $L^6$ (6.0 μmol) was treated with an equimolar mixture of $Ln^a(CF_3SO_3)_3$ and $Ln^b(CF_3SO_3)_3$ (6.0 μmol of each) in $CD_3CN/CDCl_3$ (0.6 mL) at room temperature for 1 h. The resulting yellow solution was characterized by $^1$H NMR and ESI-TOF-MS to identify the selectivity in the mixed-metal one-pot self-assembly process. No change in the $^1$H NMR spectra was observed after elongated reaction time for even 1 month, suggesting the self-assembly is fast and thermodynamically stable (Supplementary Figs. 189–194 and 289–294).

**General procedure for one-pot tri-metallic mixed-metal self-assembly of $L^1$.** $L^{1R}$ or $L^{1S}$ (6.0 μmol) was treated with a mixture of $La(ClO_4)_3 \cdot 6H_2O$, $Pr(ClO_4)_3 \cdot 6H_2O$ and $Eu(ClO_4)_3 \cdot 6H_2O$ (6.0 μmol of each) in $CD_3CN$ (0.6 mL) at 40 °C for 1 h and the turbid suspension of ligands gradually turned clear. The resulting yellow solution was characterized by $^1$H NMR and ESI-TOF-MS, which suggests a comparable self-assembly selectivity to that of two-component mixed-metal one-pot self-assembly process (Supplementary Figs. 196 and 274).

$Ca^{II}/Cd^{II}/Ln^{III}$ ($ClO_4^-$ as the counter ions) tri-metallic one-pot mixed-metal self-assembly of $L^1$ was implemented in the similar procedure as above, and

complete metal ion self-recognition was observed as ascertained by $^1$H NMR and ESI-TOF-MS (Supplementary Figs. 195 and 273).

**General procedure for post-synthetic metal-ion metathesis experiments.** Self-assembled complexes $[Ln^a_4L^1_4](ClO_4)_{12}$ (1.5 μmol) in $CD_3CN$ were prepared in advance, followed by the addition of $Ln^b(ClO_4)_3\cdot6H_2O$ (6.0 μmol), resulting in a total volume of $CD_3CN$ of 0.6 mL. $^1$H NMR spectra were measured immediately after the addition of the second $Ln^b$(III) at room temperature until the mixture reached the final thermodynamically stable state. The highly split $^1$H NMR signals and ESI-TOF-MS spectroscopy indicate the dynamic formation of multiple scrambled-metal cages $[(Ln^a_nLn^b_{4-n})L^1_4]^{12+}$ ($n = 0$–4) during the post-synthetic metal-ion metathesis experiments, which further indicates the metal-metathesis on four metal vertices of tetrahedral cages $[Ln^a_4L^1_4](ClO_4)_{12}$ proceed stepwise. It is worth pointing out that the substitution rate depends on the difference of their ionic radius in each combination. With larger ionic radii difference, the substitution proceeds much faster and vice versa. As for La(III)/Ce(III) metal combination, the one-step substitution leading to the $[La_1Ce_3L^1_4]$ species proceed so fast that no metathesis intermediates were observed (Supplementary Figs. 197–203 and 295).

Post-synthetic metal-ion metathesis experiments of $L^3$ were performed in the same method as that for $L^1$ (Supplementary Figs. 204–208).

**Single-crystal X-ray diffraction studies.** The X-ray diffraction studies for complex $Cd_4L^1_4(ClO_4)_8$ and $La_4L^1_4(ClO_4)_{12}$ were carried out at the BL17B macromolecular crystallography beamline in Shanghai Synchrotron Radiation Facility. The collected diffraction data were processed with the HKL 3000 software program[66]. The structures were solved by direct methods and refined by full-matrix least-squares on $F^2$ with anisotropic displacement using the SHELX software package[67]. The crystals diffract only very weakly due to large amounts of solvent molecules and anions. For the structure of $Cd_4L^1_4(ClO_4)_8$, where counter ions and solvent molecules were so highly disordered that they could not be reasonably located, the residual intensities were removed by PLATON/SQUEEZE routine[55]. Still one A-alert and some B-alerts are found by the (IUCr) check CIF routine, all of which are due to the poor diffraction nature of the crystals. Details on crystal data collection and refinement are summarized in Supplementary Tables 7 and 8. Additional comments on the crystallographic works are also available in the Supplementary Methods.

**Investigation of thermodynamic stability of different lanthanide complexes.** Post-synthetic metal-metathesis experiments of $Ln^{bIII}$ toward $[Ln^a_4L^1_4]^{12+}$ complexes in $CH_3CN$ were identified to proceed in a stepwise mode, with only $[Ln^a_1La^b_3L^1_4]^{12+}$ and $[Ln^b_4L^1_4]^{12+}$ complexes observed in the thermodynamic equilibrium state (for some metal combinations), as confirmed by $^1$H NMR and ESI-TOF-MS. As for $La^{III}/Ce^{III}$ combination, the displacement process is shown as below:

$$Ce + La_4L^1_4 \rightleftharpoons La_3Ce_1L^1_4 + La \qquad K_1 = \frac{[La_3Ce_1L^1_4][La]}{[La_4L^1_4][Ce]} \qquad (1)$$

$$Ce + La_3Ce_1L^1_4 \rightleftharpoons La_2Ce_2L^1_4 + La \qquad K_2 = \frac{[La_2Ce_2L^1_4][La]}{[La_3Ce_1L^1_4][Ce]} \qquad (2)$$

$$Ce + La_2Ce_2L^1_4 \rightleftharpoons La_1Ce_3L^1_4 + La \qquad K_3 = \frac{[La_1Ce_3L^1_4][La]}{[La_2Ce_2L^1_4][Ce]} \qquad (3)$$

$$Ce + La_1Ce_3L^1_4 \rightleftharpoons Ce_4L^1_4 + La \qquad K_4 = \frac{[Ce_4L^1_4][La]}{[La_1Ce_3L^1_4][Ce]} \qquad (4)$$

$$\text{Cumulative stability constant}: \beta_4 = K_1K_2K_3K_4 = \frac{[Ce_4L^1_4][La]^4}{[La_4L^1_4][Ce]^4} \qquad (5)$$

$$[La_4L^1_4] \ll [La_1Ce_3L^1_4] \rightarrow \beta_4 = \frac{[Ce_4L^1_4][La]^4}{[La_4L^1_4][Ce]^4} \gg \frac{[Ce_4L^1_4][La]^4}{[La_1Ce_3L^1_4][Ce]^4} = \beta_{La/Ce}$$

According to $^1$H NMR and ESI-TOF-MS analyses, the concentration of $[La_4L^1_4]^{12+}$ is much smaller than $[La_1Ce_3L^1_4]^{12+}$. So the minimum of $\beta_{Ce/La}$ was estimated by replacing $[La_4L^1_4]$ in Equation (5) with $[La_1Ce_3L^1_4]$, which can be defined by NMR integrations, to simplify the cumulative stability constant.

Considering that the cumulative stability constant $\beta_4$ in the metathesis experiments can be regarded as relative stability constant of $[Ce_4L^1_4]^{12+}$ toward $[La_4L^1_4]^{12+}$, $\beta_{Ce/La}$ can be used for qualitative analysis and comprehensive comparison on stability of different $[Ln_4L^1_4]^{12+}$ complexes (Supplementary Tables 4 and 5).

**Data availability.** X-ray crystal structures of compounds $La_4L^1_4$ and $Cd_4L^1_4$ reported in this paper have been deposited in the Cambridge Crystallographic Data

Center under accession numbers CCDC: 1532086 and 1532087, respectively. These data can be obtained free of charge via http://www.ccdc.cam.ac.uk/data_request/cif). All other data supporting the findings of this study are available in the article and its Supplementary Information files and from the corresponding authors on request.

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

## Acknowledgements

This work was supported by the National Natural Science Foundation of China (Grant nos.21402201, 21471150, 21521061, 21601183, 21790370, 21790374), the Strategic Priority Research Program of the Chinese Academy of Sciences (Grant No. XDB20000000), Natural Science Foundation of Fujian Province (Grant nos. 2016J06005, 2016J05051) and National Key R&D Program of China (Grant nos. 2017YFE0106900) . X.-Q.S. thanks the Science and Technology Service Network Initiative (Clean and efficient rare earth extraction and recovery technology) and "Hundreds Talents Program" from the Chinese Academy of Sciences. Q.-F.S. is grateful for the award from "The Recruitment Program of Global Youth Experts". We thank the staff of BL17B beamlines at National Center for Protein Sciences Shanghai and Shanghai Synchrotron Radiation Facility, Shanghai, People's Republic of China, for assistance during data collection.

## Author contributions

Q.-F.S. proposed the ideas and supervised the project, X.-Z.L and L.-L.Y synthesized all the compounds and conducted the experiments, L.-P. Z. contributed the mass measurements, Y.-M.D., Z.-L.B., X.-Q.S., J.D., S.W. performed the ICP-MS experiments and aided in the data analyses, X.-Z.L., J.-C.B., and Q.-F.S. wrote the manuscript with input from all the authors.

## Additional information

**Competing interests:** The authors declare no competing financial interests.

