## [Peer Review File · Nature Communications]

Reviewers' comments:

Reviewer #1 (Remarks to the Author):

Editorial Note: This Reviewer provided comments to the Editor only

Reviewer #2 (Remarks to the Author):

My apologies for the delay in responding with this report – I have found the crystallographic results fascinating but very time-consuming...

This report is concerned only with the crystallographic work reported in this paper. Both crystal samples here have presented, I suspect, huge challenges for the researchers, and provide valuable confirmation of the structures of the large M4L4 tetragonal cages. But there are few experimental details provided; I would expect to find, probably in the Supporting Information, an indication of the problems encountered and how these were overcome, for example details of the restraints and constraints applied in the refinement of such a large structure with so few intensity data. Why were the data measured at room temperature? – more, and better, data should be available at lower temperatures. Of course, I do not know what financial constraints might be involved, but I would have expected that the Synchrotron facilities used would be equipped with low-temperature devices.

Furthermore, the .CIF files for both structures are incomplete – there are many data missing. The authors should input more information about the crystal samples, the software, the structure determination and, as suggested above, especially, details of the refinement procedure.

In the main text, reference is made to the different space group symmetries found in the Cd complex versus the Eu and La complexes; are there significant differences in intermolecular contacts, counter-ion locations, etc.?

All these details should be recorded in the Supporting Information. In the main text, on line 135, I suggest that 'consistent' should be extended to 'consistent and isostructural', and a comment might be added on how similar (or not) the M4L4 cages are in the three known structures.

After the Eu4L4 structure, the Cd and La complexes may not be deemed 'novel' structures, but should nevertheless be fully reported in the Supporting Information. Additional diagrams, preferably of the ORTEP type with smaller ellipsoids, of selected individual metal centres and ligands, and a clear representation of a tetrahedral unit, should be presented here.

Overall, the crystallographic results are, I suggest, an important part of this paper, and I trust that the other components of this work have excited other referees as this part has excited me!

Reviewer #3 (Remarks to the Author):

Review of Manuscript NCOMMS-17-12226

Title: A Supramolecular Lanthanide Separation Approach Based on Multivalent Cooperative Enhancement of Metal Ion Selectivity

Authors: Xiao-Zhen Li, Li-Peng Zhou, Liang-Liang Yan, Ya-Min Dong, Zhuan-Ling Bai, Xiao-Qi Sun, Juan Diwu, Shuao Wang, Jean-Claude Bunzli, Qing-Fu Sun

Recommendation: Accept with Minor Revisions

Importance: High

Overview

This manuscript details the remarkable enhancement in metal ion selectivity of a tris-tridentate ligand through the formation of homometallic supramolecular complexes. Claims are supported by data collected from various spectroscopic and structural characterization techniques, including NMR, MS, x-ray crystallography. It is well thought out organized, and generally easy to read. This manuscript is of interest to a variety of fields not only of chemistry, but, also biology, and will likely have a large impact on future research directions.

Critique

There are grammatical and formatting errors the paper that should be addressed prior to publication. These errors and inconsistencies are not so severe that the scientific meaning of the statements are confused.

P2L55 – mixtures.

P2L58 – **there are** only a few examples of the control of lanthanide metal selectivity

P2L62 – of a multivalent

P3L107 – **Spectrum**, not spectra as only one is presented in Fig. 1b.

P4L127 – tris-tridentate

P5L129 – ~~So~~ The crystal...

P5L144 – AE is not introduced as the abbreviation for alkaline earth element until the caption in Figure 5. This is also true for TM (line 146). It is suggested that if the abbreviations are to be kept, they should be introduced when first encountered within the body of the text.

P6L161 – reacting of L¹

P6L164-167 “This absolute self-organization behavior...and ionic radii.” This is a bold statement which is based on few experimental results. I would urge the authors to be more cautious and alter the statement such that the self-organization results are **likely due to** the supramolecular cooperative mechanical coupling effect...”. There is a great deal of work which can be done in this area as follow ups to this story, and while I agree that it’s expected to be the case, without further experiments, I it should not be written as fact. The authors seem to agree with this feeling as expressed in L217-219.

P7L199 – DOSY is defined previously

P7L200 – While the mixed-metal complexes appear to be insensitive to time, there may be a barrier to change at room temperature to form the true thermodynamic product.

P8L223 – Figure 5. This is a great figure that really illustrates your observations. I do wish though that the structures of L1, L2, and L3 were introduced as figures earlier on (closer to when first introduced).

P8L227 – While studies are carried out which vary the metal and ligand ratios, it would be interesting to understand what would happen in the case where there is a 1:1 mixed metal system where the total metal to ligand ratio is 1:1. Would this induce a M_4L_4 M'_4L_4 system, or, a $M_2M'_2L_4$ arrangement? This is the case where the metals are not in large excess relative to the ligand (unlike the experiments summarized in Fig. 4).

P9L261 – Figure 6 caption incorrectly indicates that $Ln_2L^2_3$ are being compared to $Ln_4L^1_4$. The figure itself correctly indicates $Ln_3L^3_3$, however, the labels are challenging to read (they're small).

P10L264 – “indicates a ~~huge~~ **substantial** difference...”

P10L278 – “going from dinuclear to tetranuclear” The number of nuclei in these supramolecular species is greater than four. I would suggest “going from **dimetallic** to **tetrametallic**”.

P10L280 – The information on the effects of structural rigidity are very intriguing, and would be better supported with more data. Perhaps comparing the pre-organization energy of L^{1-3} with a given lanthanide would help these claims.

P11L305 – The description of the extraction experiments are concerning. While I understand that matching ligand & metal solubility is a challenge, many things can change when the solution is dried and subsequently re-dissolved. Removing the solvent may have shifted the equilibrium to form more of the ML complex than initially formed. Furthermore, the authors comment about the instability of many of the self-assembled tetrahedral complexes when exposed to water. Are the separation factors valid at all, or, are we simply observing a difference in the relative instability of the complexes. For example, perhaps the high separation factor between La(III) and Lu(III) (87.7) is because La(III) is more stable in water than Lu(III), while the low separation factor between La(III) and Pr(III) is simply because they have comparable stabilities. While this is an extreme view, I would like the authors to comment of the validity of their values as they may be at least in part due to relative rates of decomposition in water.

Reviewer #2 (Remarks to the Author):

My apologies for the delay in responding with this report – I have found the crystallographic results fascinating but very time-consuming...

Response: We thank the reviewer very much for devoting precious time to the examination of our crystallographic data and for concluding that they are fascinating.

This report is concerned only with the crystallographic work reported in this paper. Both crystal samples here have presented, I suspect, huge challenges for the researchers, and provide valuable confirmation of the structures of the large M₄L₄ tetragonal cages. But there are few experimental details provided; I would expect to find, probably in the Supporting Information, an indication of the problems encountered and how these were overcome, for example details of the restraints and constraints applied in the refinement of such a large structure with so few intensity data. Why were the data measured at room temperature? – more, and better, data should be available at lower temperatures. Of course, I do not know what financial constraints might be involved, but I would have expected that the Synchrotron facilities used would be equipped with low-temperature devices.

Response: We thank the reviewer very much for his/her understanding on the challenges of our crystallographic works and recognizing that crystallographic results provide valuable confirmations of the large M₄L₄ cage compounds. We apologize for not providing enough experimental details in the initial submission. According to his/her suggestion, we have now revised and updated all the cif files for all the X-ray data according to the suggestions. Detailed information could be found in the revised cif and checkcif files from the new submission. Main revisions are listed below:

(1) Problems encountered and how these were overcome.

Response: The main challenge in dealing with X-ray crystallography of these supramolecular complexes is to obtain enough diffraction data with reasonable resolutions, because they diffract very weakly in nature and decompose instantly after picked out from the mother liquor. Firstly, home X-ray diffractometers are used to screen the crystals and find out the best data collection conditions. After trial and error, we found out that to transfer and seal the crystals inside a glass capillary (with an atmosphere of the mother liquor without exposure to air) and collect the data quickly at room temperature gave the best results. Then, collecting the final data set on a beamline configured for biological structure determination and equipped with a high-power synchrotron X-ray source is necessary to get the best resolution data within a few minutes. Sometimes, the crystal decomposes quickly during data collection due to irradiation damage. In such a case it is necessary to merge data collected from several independent crystals.

(2) Details of the restraints and constraints.

Response: A large number of restraints and constraints have to be applied to ensure the

convergence of the refinement due to the poor data/parameter ratio. For the crystal structure of Cd₄L₄, two ligands, two Cd(II) ions, three and a half perchlorates and several hydrogen-bonded water molecules (no hydrogen was modeled in this case) are located in the asymmetrical unit. Organic ligands (“RESI 1” and “RESI 2”) and perchlorates (“RESI 3”) are separately labeled under the same scheme and are forced to adopt similar configurations as restrained by the “SAME” commands. For the ligand structure, “AFIX 66” constraints have been applied to the six-membered aromatic rings including pyridines. Moreover, many geometrical restraints including “FLAT”, “DFIX”, “DANG”, “SADI” are applied to the ligands and the perchlorate ions based on the X-ray coordinates of a similar Eu₄L₄ structure from our previous work (*J. Am. Chem. Soc.* **2015**, 137, 8550–8555; CCDC-1057536), which was determined to a much better resolution. For more detailed information of these geometrical restraints, please allow us to direct the referee to the final .CIF files, where the full _shelx_res_files have now been incorporated.

Because of the restraints and constraints used, we have to emphasize that the main purpose of providing the X-ray structures is in general merely to confirm the connectivity of the target assemblies. For this reason, discussions based on the crystal structures are always kept to the minimum.

(3) Why were the data measured at room temperature?

Response: X-ray data collection at cryogenic conditions for these compounds resulted in the deterioration of crystallinity due to unknown reasons and gave worse quality of data. As mentioned above, after trial and error screening, we found out that to transfer and seal the crystals inside a glass capillary (with an atmosphere of the mother liquor without exposure to air) and collect the data quickly at room temperature gave the best results. The reason for the deterioration of the crystals under cryogenic conditions possibly has something to do with the large cavities existing inside the big unit cells that are filled with amorphous organic solvents such as diethyl ether, THF and so on, which may still slowly diffuse/evaporate under liquid N₂ temperature. This is a quite different feature from biological samples which are always grown from water. So it is quite common to see X-ray data collected at room temperature for supramolecular systems. Our own experiences (for example: *Science*, **2010**, 328, 1144-1147; *Nature Chem.*, **2012**, 4, 330-333; *J. Am. Chem. Soc.*, **2015**, 137, 8550–8555) also suggest that to seal the crystals inside a glass capillary and collect the data quickly at room temperature may be a general protocol to follow for fragile crystals grown from volatile organic solvents.

Furthermore, the .CIF files for both structures are incomplete – there are many data missing. The authors should input more information about the crystal samples, the software, the structure determination and, as suggested above, especially, details of the refinement procedure.

Response: We thank the reviewer for the kind suggestions. All the details pointed out by the reviewer have now been introduced into the revised CIF files.

In the main text, reference is made to the different space group symmetries found in the Cd complex versus the Eu and La complexes; are there significant differences in intermolecular contacts,

counter-ion locations, etc.? All these details should be recorded in the Supporting Information. In the main text, on line 135, I suggest that 'consistent' should be extended to 'consistent and isostructural', and a comment might be added on how similar (or not) the M₄L₄ cages are in the three known structures.

Response: We thank the reviewer for the kind suggestions. Indeed, although the discrete M₄L₄-type tetrahedral complexes are 'isostructural' in nature, there are some distinct differences between Cd complex and La complex in their packing diagrams in the crystal states. For example, Cd₄(L¹)₄ tetrahedral cages are very loosely packed in the ab plane and there are infinite channels with diameters of ca. 2.24 nm along the c axis. As a result, 'void' occupancy as much as 50.6% is calculated in the unit cell based on PLATON. In a clear contrast, La₄(L¹)₄ tetrahedral cages are much densely packed and only 32.5% 'void' occupancy is found by PLATON (Figure 37). As a result, crystals of Cd₄(L¹)₄ diffract much weakly than those of La₄(L¹)₄. In the main text, on line 135, 'consistent' has been changed to 'consistent and isostructural' and a short comment as mentioned above has been added to the main text as suggested by the reviewer.

Figure 37. Different crystal packing diagrams for the Cd₄L¹₄ (left) and Ln₄L¹₄ (right) tetrahedral cages viewing along the c axes.

After the Eu₄L₄ structure, the Cd and La complexes may not be deemed 'novel' structures, but should nevertheless be fully reported in the Supporting Information. Additional diagrams, preferably of the ORTEP type with smaller ellipsoids, of selected individual metal centres and ligands, and a clear representation of a tetrahedral unit, should presented here.

Response: We thank the reviewer for the suggestions. The following figures have now been added to the SI.

Figure 38. ORTEP drawing of the individual metal centres and ligands in the asymmetrical unit of Cd_4L_{14} at 30% ellipsoids level.

Figure 39. ORTEP drawing of the Cd_4L_{14} tetrahedral complex at 30% ellipsoids level.

Figure 40. ORTEP drawing of the individual metal centres and ligands in the asymmetrical unit of La_4L_{14} at 30% ellipsoids level.

Figure 41. ORTEP drawing of the La₄L₄ tetrahedral complex at 30% ellipsoids level.

Overall, the crystallographic results are, I suggest, an important part of this paper, and I trust that the other components of this work have excited other referees as this part has excited me!

Response: We thank this reviewer very much for his/her very kind appreciation of our work.

Reviewer #3 (Remarks to the Author):

Overview

This manuscript details the remarkable enhancement in metal ion selectivity of a tris-tridentate ligand through the formation of homometallic supramolecular complexes. Claims are supported by data collected from various spectroscopic and structural characterization techniques, including NMR, MS, xray crystallography. It is well thought out organized, and generally easy to read. This manuscript is of interest to a variety of fields not only of chemistry, but, also biology, and will likely have a large impact on future research directions.

Response: We thank this reviewer very much for his/her kind appreciation of our work.

Critique

There are grammatical and formatting errors the paper that should be addressed prior to publication. These errors and inconsistencies are not so severe that the scientific meaning of the statements are confused.

Response: We thank this reviewer very much for his/her very thorough reading on our manuscript. All the language problems have now been fixed, in line with the reviewer's suggestions.

P2L55 – mixtures.

Response: mixtures.

*P3L58 – **there are** only a few examples of the control of lanthanide metal selectivity*

Response: there are only a few examples

P2L62 – of a multivalent

Response: of a multivalent

*P3L107 – **Spectrum**, not spectra as only one is presented in Fig. 1b.*

Response: spectrum

P4L127 – tris-tridentate

Response: tris-tridentate

*P5L129 – So **The** crystal...*

Response: So the crystal

P5L144 – AE is not introduced as the abbreviation for alkaline earth element until the caption in Figure 5. This is also true for TM (line 146). It is suggested that if the abbreviations are to be kept, they should be introduced when first encountered within the body of the text.

Response: Introduction of these abbreviations has been added in the main text.

P6L161 – reacting of L1

Response: reaction of L¹

*P6L164-167 “This absolute self-organization behavior...and ionic radii.” This is a bold statement which is based on few experimental results. I would urge the authors to be more cautious and alter the statement such that the self-organization results are **likely due to** the supramolecular cooperative mechanical coupling effect...”. There is a great deal of work which can be done in this area as follow ups to this story, and while I agree that it’s expected to be the case, without further experiments, I it should not be written as fact. The authors seem to agree with this feeling as expressed in L217-219.*

Response: These statements have been modified.

P7L199 – DOSY is defined previously

Response: The repeated sentence has been deleted.

P7L200 – While the mixed-metal complexes appear to be insensitive to time, there may be a barrier to change at room temperature to form the true thermodynamic product.

Response: ¹H NMR spectrum of La^{III}/Ce^{III} mixed-metal self-assembled complexes with L¹ showed no difference after reacting at 65°C for 2 days (Figure 42). This suggests that the assembled complexes we obtained at 65°C are thermodynamic products.

Figure 42. ^1H NMR spectra (400 MHz, CD_3CN , 298K) of $\text{La}^{\text{III}}/\text{Ce}^{\text{III}}$ mixed-metal self-assembled complexes with L^1 (CF_3SO_3^- salt).

P8L223 – Figure 5. This is a great figure that really illustrates your observations. I do wish though that the structures of L1, L2, and L3 were introduced as figures earlier on (closer to when first introduced).

Response: The structure of ligand L^1 is given in Figure 5 at the beginning of the “Result” section. Although briefly mentioned in the introduction, properties of complexes with ligands L^2 and L^3 start only to be described in Figure 5; therefore we prefer not to change.

P8L227 – While studies are carried out which vary the metal and ligand ratios, it would be interesting to understand what would happen in the case where there is a 1:1 mixed metal system where the total metal to ligand ratio is 1:1. Would this induce a $\text{M}4\text{L}4/\text{M}'4\text{L}4$ system, or, a $\text{M}2\text{M}'2\text{L}4$ arrangement? This is the case where the metals are not in large excess relative to the ligand (unlike the experiments summarized in Fig. 4).

Response: We thank the reviewer for suggesting this interesting control experiment. Three representative lanthanide pairs $\text{Lna}^{\text{III}}/\text{Lnb}^{\text{III}}$ ($\text{La}^{\text{III}}/\text{Nd}^{\text{III}}$, $\text{La}^{\text{III}}/\text{Eu}^{\text{III}}$, $\text{La}^{\text{III}}/\text{Lu}^{\text{III}}$) are used to figure out the metal ion distribution in final assembly where total metal ions to ligand ratio is 1:1 ($\text{Ln}_a^{\text{III}}/\text{Ln}_b^{\text{III}}/\text{L}^1 = 0.5/0.5/1$). In theory, pure statistically-distributed mixtures of $[\text{Ln}_n\text{Ln}_{4-n}(\text{L}^1)_4]^{12+}$ ($n = 0-4$) species would favour the formation of the $[\text{Ln}_2\text{Ln}_2(\text{L}^1)_4]^{12+}$ complex (statistical distribution ratio: $[\text{Ln}_4(\text{L}^1)_4]^{12+} : [\text{Ln}_3\text{Ln}_1(\text{L}^1)_4]^{12+} : [\text{Ln}_2\text{Ln}_2(\text{L}^1)_4]^{12+} : [\text{Ln}_1\text{Ln}_3(\text{L}^1)_4]^{12+} : [\text{Ln}_4(\text{L}^1)_4]^{12+} = \text{C}_4^4 : \text{C}_4^1 \text{C}_4^3 : \text{C}_4^2 \text{C}_4^2 : \text{C}_4^3 \text{C}_4^1 : \text{C}_4^4 = 1 : 16 : 36 : 16 : 1$). However, ^1H NMR spectra and ESI-TOF-MS indicated the biased formation of two

homometallic cages, along with some statistical mixture of heterometallic complexes (Figures 43-50). For lanthanide pair of $\text{La}^{\text{III}}/\text{Nd}^{\text{III}}$, nonlinear curve fitting of the isotope patterns using the following models, a : $[\text{La}_4(\text{L}^1)_4]^{12+}$, b: $[\text{La}_3\text{Nd}_1(\text{L}^1)_4]^{12+}$, c: $[\text{La}_2\text{Nd}_2(\text{L}^1)_4]^{12+}$, d: $[\text{La}_1\text{Nd}_3(\text{L}^1)_4]^{12+}$, e: $[\text{Nd}_4(\text{L}^1)_4]^{12+}$ ($4a+3b+2c+d=b+2c+3d+4e$) has been carried out. It turns out that the observed isotope patterns approximately agrees with a composition of $[\text{La}_4(\text{L}^1)_4]^{12+} : [\text{La}_3\text{Nd}_1(\text{L}^1)_4]^{12+} : [\text{La}_2\text{Nd}_2(\text{L}^1)_4]^{12+} : [\text{La}_1\text{Nd}_3(\text{L}^1)_4]^{12+} : [\text{Nd}_4(\text{L}^1)_4]^{12+} = 1.08 : 1.94 : 1.28 : 1 : 1.55$ (Figures 43-45). As for lanthanide pairs of $\text{La}^{\text{III}}/\text{Eu}^{\text{III}}$ and $\text{La}^{\text{III}}/\text{Lu}^{\text{III}}$, higher proportion of homometallic complexes $[\text{Lna}_4(\text{L}^1)_4]^{12+}$ and $[\text{Lnb}_4(\text{L}^1)_4]^{12+}$ was also achieved (Figures 46-50). Though no pure narcissistic self-sorting happened in these systems, such an observation nevertheless demonstrates the power of metal ion selectivity of our ligand.

Figure 43. ^1H NMR spectra (400 MHz, CD_3CN , 298K) of $\text{La}^{\text{III}}/\text{Nd}^{\text{III}}$ mixed-metal self-assembled complexes with L^1 (CF_3SO_3^- salt) with stoichiometric ratio $\text{La}^{\text{III}} : \text{Nd}^{\text{III}} : \text{L}^1 = 0.5 : 0.5 : 1$.

Figure 44. ESI-Q-TOF mass spectrum of $\text{La}^{\text{III}}/\text{Nd}^{\text{III}}$ mixed-metal self-assembled complexes

with L^1 (ClO_4^- salt) with the metal and ligand ratio as $\text{La}^{\text{III}} : \text{Nd}^{\text{III}} : L^1 = 0.5 : 0.5 : 1$.

Figure 45. Nonlinear curve fitting of simulated isotope patterns of $\text{La}^{\text{III}}/\text{Nd}^{\text{III}}$ mixed-metal self-assembled complexes with L^1 (ClO_4^- salt) with the metal and ligand ratio as $\text{La}^{\text{III}} : \text{Nd}^{\text{III}} : L^1 = 0.5 : 0.5 : 1$.

Nonlinear curve fitting of the isotope patterns of $\text{La}^{\text{III}}/\text{Nd}^{\text{III}}$ mixed-metal self-assembled complexes with L^1 (ClO_4^- salt) results in a composition of $[\text{La}_4(\text{L}^1)_4]^{12+} : [\text{La}_3\text{Nd}_1(\text{L}^1)_4]^{12+} : [\text{La}_2\text{Nd}_2(\text{L}^1)_4]^{12+} : [\text{La}_1\text{Nd}_3(\text{L}^1)_4]^{12+} : [\text{Nd}_4(\text{L}^1)_4]^{12+} = 1.08 : 1.94 : 1.28 : 1 : 1.55$ (red dash).

Figure 46. Simulated isotope patterns of pure statistically-distributed mixtures of $\text{La}^{\text{III}}/\text{Nd}^{\text{III}}$ mixed-metal self-assembled complexes with L^1 (ClO_4^- salt) with the metal and ligand ratio as $\text{La}^{\text{III}} : \text{Nd}^{\text{III}} : L^1 = 0.5 : 0.5 : 1$.

The statistical distribution ratio of mixtures of $[\text{La}_n\text{Lnb}_{4-n}(\text{L}^1)_4]^{12+}$ ($n = 0-4$) species:
 $[\text{La}_4(\text{L}^1)_4]^{12+} : [\text{La}_3\text{Lnb}_1(\text{L}^1)_4]^{12+} : [\text{La}_2\text{Lnb}_2(\text{L}^1)_4]^{12+} : [\text{La}_1\text{Lnb}_3(\text{L}^1)_4]^{12+} : [\text{Lnb}_4(\text{L}^1)_4]^{12+} = C_4^4 : C_4^1 C_4^3 : C_4^2 C_4^2 : C_4^3 C_4^1 : C_4^4 = 1 : 16 : 36 : 16 : 1$.

Figure 47. ^1H NMR spectra (400 MHz, CD_3CN , 298K) of $\text{La}^{\text{III}}/\text{Eu}^{\text{III}}$ mixed-metal self-assembly complexes with L^1 (CF_3SO_3^- salt) with the metal and ligand ratio as $\text{La}^{\text{III}} : \text{Eu}^{\text{III}} : \text{L}^1 = 0.5 : 0.5 : 1$.

Figure 48. Simulated isotope patterns of pure statistically-distributed mixtures of $\text{La}^{\text{III}}/\text{Eu}^{\text{III}}$ mixed-metal self-assembly complexes with L^1 (CF_3SO_3^- salt) with the metal and ligand ratio as $\text{La}^{\text{III}} : \text{Eu}^{\text{III}} : \text{L}^1 = 0.5 : 0.5 : 1$.

Figure 49. ^1H NMR spectra (400 MHz, CD_3CN , 298K) of $\text{La}^{\text{III}}/\text{Lu}^{\text{III}}$ mixed-metal self-assembly complexes with L^1 (CF_3SO_3^- salt) with the metal and ligand ratio as $\text{La}^{\text{III}} : \text{Lu}^{\text{III}} : \text{L}^1 = 0.5 : 0.5 : 1$.

Figure 50. Simulated isotope patterns of pure statistically-distributed mixtures of $\text{La}^{\text{III}}/\text{Lu}^{\text{III}}$ mixed-metal self-assembly complexes with L^1 (CF_3SO_3^- salt) with the metal and ligand ratio as $\text{La}^{\text{III}} : \text{Lu}^{\text{III}} : \text{L}^1 = 0.5 : 0.5 : 1$.

P9L261 – Figure 6 caption incorrectly indicates that Ln2L23 are being compared to Ln4L14. The figure itself correctly indicates Ln3L3 3, however, the labels are challenging to read (they're small).

Response: Labels in Figure 6 in the main text have been enlarged and figure caption has been corrected

P10L264 – “indicates a huge **substantial** difference...”

Response: “indicates a substantial difference...”

P10L278 – “going from dinuclear to tetranuclear” The number of nuclei in these supramolecular species is greater than four. I would suggest “going from **dimetallic** to **tetrametallic**”.

Response: “going from dimetallic to tetrametallic”.

P10L280 – The information on the effects of structural rigidity are very intriguing, and would be better supported with more data. Perhaps comparing the pre-organization energy of L1-3 with a given lanthanide would help these claims.

Response: We thank the reviewer very much for drawing our attention to the calculation of pre-organization energies in our system. According to literature method reported by Piguet et al (*Chem. Commun.*, **2010**, 46, 6209–6231; *Chem. Eur. J.* **2005**, 11, 5217 – 5226; *Chem. Eur. J.* **2005**, 11, 5227 – 5237), the thermodynamic self-assembly of any metallo-supramolecular complex [M_{pm}L_{pn}] in solution can be dissected into five additive free energy contributions.

Symmetry	C ₃	D _{3h}	T	C _{3v}
σ ^{ext}	3	6	12	3
σ ^{int}	3 ³	3 ⁹	1	1
σ ^{chiral}	1/2	1	1/2	1

$$\beta_{4,4}^{\text{Eu,L}^1} = \omega_{4,4}^{\text{chiral}} \omega_{4,4}^{\text{Eu,L}^1} (f_{\text{inter}})^7 (f_{\text{intra}})^9 = \omega_{4,4}^{\text{chiral}} \omega_{4,4}^{\text{Eu,L}^1} (f_{\text{inter}})^{16} (\text{EM}_{4,4}^{\text{Eu,L}^1})^9 (u^{\text{L}^1,\text{L}^1})^{12} (u^{\text{Eu,Eu}})^6$$

$$= 581130733.5 (f_{\text{inter}})^{16} (\text{EM}_{4,4}^{\text{Eu,L}^1})^9 (u^{\text{L}^1,\text{L}^1})^{12} (u^{\text{Eu,Eu}})^6 \quad (1)$$

Symmetry	C ₂	D _{3h}	D ₃	C _{3v}
σ ^{ext}	2	6	6	3
σ ^{int}	3 ²	3 ⁹	1	1
σ ^{chiral}	1/2	1	1/2	1

$$\beta_{2,3}^{\text{Eu,L3}} = \omega_{2,3}^{\text{chiral}} \omega_{2,3}^{\text{Eu,L3}} (f_{\text{inter}})^4 (f_{\text{intra}})^2 = \omega_{2,3}^{\text{chiral}} \omega_{2,3}^{\text{Eu,L3}} (f_{\text{inter}})^6 (EM_{2,3}^{\text{Eu,L3}})^2 (u^{\text{L1,L1}})^6 (u^{\text{Eu,Eu}})^6$$

$$= 8748 (f_{\text{inter}})^6 (EM_{2,3}^{\text{Eu,L3}})^2 (u^{\text{L1,L1}})^6 (u^{\text{Eu,Eu}})^6 \quad (2)$$

Where $\beta_{\text{pm,pn}}^{\text{M,L}}$ is the associated cumulative stability constant, f_{inter} and f_{intra} correspond to the absolute affinities of the binding sites for the intermolecular and intramolecular connections with the metal ion in terms of stability constants, $\omega_{\text{pm,pn}}^{\text{M,L}}$ takes into account the pure statistical contribution due to the change in the molecular rotational degeneracies occurring when the reactants are transformed into products and is calculated using the symmetry numbers σ of each partner. EM is the effective concentration that corrects f_{inter} for intramolecular macrocyclic complexation processes and $u^{\text{M,M}} = \exp(-\Delta E^{\text{M,M}}/RT)$ and $u^{\text{L,L}} = \exp(-\Delta E^{\text{L,L}}/RT)$ are Boltzmann factors that account for the intermetallic $\Delta E^{\text{M,M}}$ and interligand $\Delta E^{\text{L,L}}$ interactions.

The empirical experimental parameter EM, termed the effective molarity (because of its formal concentration units) has been extensively used as a measure of the ease of intramolecular connection with respect to the alternative intermolecular process (*Dalton Trans.*, **2006**, 0, 1473–1490).

$$\Delta G_{\text{preorg}} = -RT \ln(EM)$$

The free energies of pre-organization corresponding to the correction which applies when the intermolecular process is replaced by the intramolecular one

$$EM = K_{\text{intra}}/K_{\text{inter}}$$

$$-RT \ln(EM) = -RT \ln(K_{\text{intra}}) + RT \ln(K_{\text{inter}}) = \Delta G_{\text{intra}} - \Delta G_{\text{inter}}$$

The introduction of enthalpic and entropic contributions to each type of connection and the hypothesis of formation of strainless ring result in:

$$-RT \ln(EM) = (\Delta H_{\text{intra}} - H_{\text{inter}}) - T(\Delta S_{\text{intra}} - \Delta S_{\text{inter}}) \approx -T(\Delta S_{\text{intra}} - \Delta S_{\text{inter}})$$

$$EM = \exp((\Delta S_{\text{intra}} - \Delta S_{\text{inter}})/R)$$

The four parameters in equation (1) and (2) can be fitted (nonlinear least-squares fit) through at least five independent macroscopic constants that characterize comparable complexation processes. The pre-organization energy of L^{1-3} can be compared through EM. However, lack of intermediate complexes in the existence of excess ligands L^1 and derived complexes in the existence of excess metal ions leads to deficiency of experimental stability constants and prevent the calculation of EM.

EM can also be roughly estimated through Kuhn theory, the well-accepted EM/a^{-3} dependence, in which a is the separation between the two binding sites. But this can be applied to models that have only two coordination sites, i.e. bidentate ligands (*Chem. Eur. J.* **2008**, 14, 2994 – 3005). Clearly, the situation here is much more complicated with the tris-tridentate ligands that form tetrahedral complexes.

Concerning the difficulty in calculating the pre-organization energy of L^1 , ligand competition experiment was performed to compare the stability of $[(\text{Eu})_4(\text{L}^1)_4]^{12+}$ and $[(\text{Eu})_2(\text{L}^3)_3]^{6+}$. A mixture of L^1 and L^3 (2.5 eq and 3.5 eq, respectively) was prepared and reacted with Eu^{III} (2 eq), resulting in the formation of $[(\text{Eu})_4(\text{L}^1)_4]^{12+}$ as the main species after 36h, as demonstrated by ^1H NMR and ESI-TOF-MS (Figures 51,52). Only very weak signals due to $[(\text{Eu})_2(\text{L}^3)_3]^{6+}$

are detected, in addition to small signals from the free ligand L^1 . This selective self-assembly of Eu^{III} with tris-tridentate ligand L^1 suggests a larger formation constant of $[(\text{Eu})_4(\text{L}^1)_4]^{12+}$ compared to $[(\text{Eu})_2(\text{L}^3)_3]^{6+}$. Both L^1 and L^3 are in excess with respect to Eu^{III} in this experiment to exclude the possibility that the formation of one species is induced by the residual metal ions after complexing with the other ligand. Large amount of free ligand L^3 exists in the reaction system since the preferential self-assembly of Eu^{III} with L^1 . But no signals of L^3 were observed in the ^1H NMR spectra because of its rather low solubility in CD_3CN .

Figure 51. ^1H NMR spectra (400 MHz, CD_3CN , 298K) of L^1/L^3 mixed-ligand self-assembly with Eu^{III} (CF_3SO_3^- salt).

Figure 52. ESI-Q-TOF mass spectrum of L¹/L³ mixed-ligand self-assembled complexes with Eu^{III} (CF₃SO₃⁻ salt). Only small amount of [Eu₂L₃]⁶⁺ was observed with its peaks labelled with orange triangle.

P11L305 – The description of the extraction experiments are concerning. While I understand that matching ligand & metal solubility is a challenge, many things can change when the solution is dried and subsequently re-dissolved. Removing the solvent may have shifted the equilibrium to form more of the ML complex than initially formed. Furthermore, the authors comment about the instability of many of the self-assembled tetrahedral complexes when exposed to water. Are the separation factors valid at all, or, are we simply observing a difference in the relative instability of the complexes. For example, perhaps the high separation factor between La(III) and Lu(III) (87.7) is because La(III) is more stable in water than Lu(III), while the low separation factor between La(III) and Pr(III) is simply because they have comparable stabilities. While this is an extreme view, I would like the authors to comment of the validity of their values as they may be at least in part due to relative rates of decomposition in water.

Response: We thank the reviewer for pointing out the stability concerns in our extraction experiments. In order to exclude the probability that the high separation factor in the extraction process may profit from the higher stability of [(L_a)₄(L¹)₄]¹²⁺ with respect to [(L_b)₄(L¹)₄]¹²⁺ in water, selectivity experiments have been performed again in CD₃CN containing 10% (v/v) D₂O. ¹H NMR spectra showed no difference with those measured in pure CD₃CN (Figure 53-57). Note that the proton signals of the amide groups disappeared due to H/D exchange with D₂O. These results indicate that relative rates of decomposition in water do not contribute to the high separation factor. Moreover, good validity of the separation efficiency has been also approved through parallel extraction experiments and the results are listed in Table 4.

Figure 53. ¹H NMR spectra (400 MHz, CD₃CN, 298K) of La^{III}/Pr^{III} mixed-metal self-assembled complexes with L¹ (CF₃SO₃⁻ salt).

Figure 54. ¹H NMR spectra (400 MHz, CD₃CN, 298K) of Pr^{III}/Eu^{III} mixed-metal self-assembled complexes with L¹ (CF₃SO₃⁻ salt).

Figure 55. ¹H NMR spectra (400 MHz, 298K) of La^{III}/Eu^{III} mixed-metal self-assembled complexes with L¹ (CF₃SO₃⁻ salt).

Figure 56. ¹H NMR spectra (400 MHz, 298K) of La^{III}/Lu^{III} mixed-metal self-assembled complexes with L¹ (CF₃SO₃⁻ salt).

Figure 57. ^1H NMR spectra (400 MHz, 298K) of $\text{Eu}^{\text{III}}/\text{Lu}^{\text{III}}$ mixed-metal self-assembled complexes with L^1 (CF_3SO_3^- salt).

Table 4. Extraction performance of L^6 self-assembled complexes in $\text{CHCl}_3/\text{H}_2\text{O}$.

Metal combination	Starting material/mg	Extract in water/mg	$S_{\text{Lna/Lnb}}$
La/Pr	1.18/1.21	0.37/0.219	2.1
	1.28/1.29	0.5972/0.3637	2.2
	1.32/1.24	0.5614/0.3367	1.9
La/Eu	1.00/1.07	0.49/0.11	8.4
	1.19/1.33	0.154/0.022	8.9
	1.25/1.34	0.1418/0.0207	8.1
La/Lu	1.17/1.40	0.57/0.015	87.7
	1.22/1.49	0.8543/0.039	86.7
	1.33/1.61	0.54/0.013	84.0
Pr/Eu	1.10/1.24	0.32/0.127	3.6
	1.29/1.37	0.0838/0.0264	3.6
	1.30/1.36	0.0715/0.0192	4.1
Eu/Lu	1.26/1.36	0.12/0.015	9.4
	1.34/1.54	0.5592/0.0976	10.6
	1.36/1.55	0.477/0.1018	7.7

REVIEWERS' COMMENTS:

Reviewer #1 (Remarks to the Author):

This manuscript describes an analysis of the metal binding selectivity found upon treatment of rare earth metals (and some other main group/transition metals) with a tris-tridentate coordinating ligand. The selectivity is impressive, and surprising, given the lack of selectivity usually found when neutral tris-coordinate ligands are used to bind rare earth metals. Very small selectivities are found for other ligands that use this type of C=O, pyridyl-, C=O coordinator. Even though there are 4 possibilities for incorporating different metals in the complex, almost narcissistic formation of cages is seen in this case.

The novelty, and the impact, lies in the selectivity of assembly of the Ln complexes. As the authors note, the selectivity is unprecedented, and is certainly the most impressive example of Ln-based metal-selective binding and sorting to date. The ESI-MS data is impressively detailed and persuasive, and the NMR data has been revised and justified well in this resubmission. The authors have done a lot of work to show that the (surprising) selectivity is consistent and repeatable. I still don't know why this works, and I am looking forward to the authors' next paper - I'm sure there are lots of studies to be done to tease out the origins of the selectivity. As an initial communication, this is worthy of publication in Nature Communications - the observation is novel and exciting, the corrections by the authors are detailed and extensive, and the data is clean.

Reviewer #2 (Remarks to the Author):

Further comments on the crystallography in this paper:

I am pleased with the extra details supplied by the authors; the structure analyses were, indeed, as awkward as I imagined, and it is good to have these details in print, if mainly in the Supporting Information.

I am not so sure about the extra ORTEP diagrams – they are extremely complicated! – and I cannot see how they can easily be made clearer. I shall leave that choice to the authors and editors!

Most of my further queries have been answered in great detail and included in the Supporting Information and in the .CIF files – thank you.

I look forward to seeing this paper proceeding to publication.

Reviewer #3 (Remarks to the Author):

The authors have adequately addressed all comments and concerns regarding their work in this manuscript.

The authors provide strong evidence for their conclusions. The results and work they present are novel, of great importance to scientists in a variety of fields, and will be of interest to researchers in related disciplines.

My recommendation is that this manuscript be accepted for publication.

Spelling and grammar corrections:

Line 260 (page 9) - predominant -> predominant

Line 306 (page 11) - aminomethy -> aminomethyl

Response to the reviewers

For the comments of Reviewer 1:

Reviewer #1 (Remarks to the Author):

Comments:

This manuscript describes an analysis of the metal binding selectivity found upon treatment of rare earth metals (and some other main group/transition metals) with a tris-tridentate coordinating ligand. The selectivity is impressive, and surprising, given the lack of selectivity usually found when neutral tris-coordinate ligands are used to bind rare earth metals. Very small selectivities are found for other ligands that use this type of C=O, pyridyl-, C=O coordinator. Even though there are 4 possibilities for incorporating different metals in the complex, almost narcissistic formation of cages is seen in this case.

The novelty, and the impact, lies in the selectivity of assembly of the Ln complexes. As the authors note, the selectivity is unprecedented, and is certainly the most impressive example of Ln-based metal-selective binding and sorting to date. The ESI-MS data is impressively detailed and persuasive, and the NMR data has been revised and justified well in this resubmission. The authors have done a lot of work to show that the (surprising) selectivity is consistent and repeatable. I still don't know why this works, and I am looking forward to the authors' next paper - I'm sure there are lots of studies to be done to tease out the origins of the selectivity. As an initial communication, this is worthy of publication in Nature Communications - the observation is novel and exciting, the corrections by the authors are detailed and extensive, and the data is clean.

Response: We thank the reviewer very much for the kind appreciation of our work and the really good advice to our future plans.

For the comments of Reviewer 2:

Reviewer #2 (Remarks to the Author):

Further comments on the crystallography in this paper:

I am pleased with the extra details supplied by the authors; the structure analyses were, indeed, as awkward as I imagined, and it is good to have these details in print, if mainly in the Supporting Information.

I am not so sure about the extra ORTEP diagrams – they are extremely complicated! – and I cannot see how they can easily be made clearer. I shall leave that choice to the authors and editors!

Most of my further queries have been answered in great detail and included in the Supporting Information and in the .CIF files – thank you.

I look forward to seeing this paper proceeding to publication.

Response: We thank the reviewer very much for the patient and prudential examination of our crystallographic data. We have added the detailed structure analyses in the manuscript and

Supporting Information to facilitate a better understanding of the crystallography in this work. Moreover, the details of our explanations appeared in the rebuttal letter will also be published.

For the comments of Reviewer 3:

Reviewer #3 (Remarks to the Author):

The authors have adequately addressed all comments and concerns regarding their work in this manuscript.

The authors provide strong evidence for their conclusions. The results and work they present are novel, of great importance to scientists in a variety of fields, and will be of interest to researchers in related disciplines.

My recommendation is that this manuscript be accepted for publication.

Spelling and grammar corrections:

Line 260 (page 9) - predominant -> predominant

Line 306 (page 11) - aminomethy -> aminomethyl

Response: We thank the reviewer very much for his/her kind appreciation and the valuable suggestions to our work. The spelling and grammar mistakes have been corrected.